# Urban vacant lands impart hydrological benefits across city landscapes

Christa Kelleher [1✉], Heather E. Golden [2], Sean Burkholder[3] & William Shuster [2,4]

Cities evolve through phases of construction, demolition, vacancy, and redevelopment, each impacting water movement at the land surface by altering soil hydrologic properties, land cover, and topography. Currently unknown is whether the variable physical and vegetative characteristics associated with vacant parcels and introduced by demolition may absorb rainfall and thereby diminish stormwater runoff. To investigate this, we evaluate how vacant lots modulate citywide hydrologic partitioning by synthesizing a novel field dataset across 500+ parcels in Buffalo, New York, USA. Vacant lot infiltration rates vary widely (0.001 to $5.39\,cm\,h^{-1}$), though parcels are generally well-vegetated and gently sloped. Extending field estimates to 2400 vacant parcels, we estimate that vacant lands citywide may cumulatively infiltrate 51–54% additional annual rainfall volume as compared to pre-demolition state, in part by reducing and disconnecting impervious areas. Our findings differentiate vacant lots as purposeful landscapes that can alleviate large water fluxes into aging wastewater infrastructure.

[1] Department of Earth Sciences and Civil Engineering, Syracuse University, 141 Crouse Dr, Syracuse, New York, NY 13244, USA. [2] Office of Research and Development, U.S. Environmental Protection Agency, 26 Martin Luther King Dr W, Cincinnati, OH 45268, USA. [3] Department of Landscape Architecture, University of Pennsylvania, Meyerson Hall, 210S 34th St #119, Philadelphia, PA 19104, USA. [4] Department of Civil and Environmental Engineering, Wayne State University, 5050 Anthony Wayne Dr, Detroit, MI 48202, USA. ✉email: ckellehe@syr.edu

Urban landscapes across the globe are constantly changing via dynamic cycles of development, vacancy, and redevelopment, which in turn drives urban environmental change[1–5]. As a result of economic forces (e.g., the United States mortgage crisis, ca. 2008[6,7]), low land values, and efforts to control blight through demolition, the extent of vacant land has recently and rapidly expanded in many cities[5,8,9]. Estimates show that vacant land can constitute more than 15% of the total land area in cities throughout the US[10,11]. Pervasive blight and the associated decline in residential housing values have driven demolition in many cities, creating more vacant land with uncertain benefits to cities and their residents[12–14].

In recent years, the once negative perception of vacant parcels has begun to shift as vacant lands are prioritized for redevelopment to preserve green space and, in some cases, to render stormwater management services[15–18]. From a hydrologic perspective, these parcel-scale, pervious vacant lots can accumulate and intersperse with existing impervious surfaces to create novel arrangements of green space[19–23]. Furthermore, these mosaic landscapes can potentially offer different degrees of rainfall detention capacity (RDT), which may provide hydrologic benefits[17,24,25]. Here, we define RDT as the volume of infiltrated water (i.e., the product of infiltration rate and pervious parcel lot area) for a given rainfall event. In addition to rainfall rate, detention capacity will be modified by surface landform and soil properties, which present as highly variable across small (e.g., parcels) to large (e.g., cities) spatial scales[17,24].

In order to maintain reliable wastewater services to residents, cities seek sustainable, cost-effective approaches to manage urban runoff and system conveyance capacity[4,26,27]. Historically, green infrastructure (GI; the use of soils, plants, and landscape design to control stormwater[28,29]) has been introduced across many urban areas as a decentralized, distributed approach to reduce sewer system overflows and to deliver other tangible benefits[30–34]. Overall, GI often aims to regulate water volume inputs into the system and thereby reduce the frequency and magnitude of, for example, combined sewer overflows (CSOs)—which many cities are currently under orders to reduce. In particular, when the volume of combined or separated sewage and stormwater exceeds system capacity in a sewered catchment area (the sewershed), conveyance systems overflow and transfer this excess volume to nearby surface waters. This has resulted in deleterious water quality and human health impacts in municipalities around the world[3,35,36].

An emergent management strategy in the USA is to employ permeable vacant lands as a way to reduce the generation of excess stormwater runoff[37,38]. This potential runoff is detained in the vacant parcels and kept out of the wastewater collection and conveyance system. In this way, vacant land may be used to reduce the frequency and volume of CSOs, and help fulfill the objectives of federally-mandated consent orders and their associated long-term control plans. This strategy is integrated into the demolition process, which creates vacant land, transforming formerly impervious to pervious surfaces that can be used as a structural tool in stormwater management.

In this vein, we posit that vacant lands may represent an opportunity for managing urban stormwater in cities with large portfolios of permeable vacant parcels that are decentralized throughout a city (and its wastewater service areas)[9,15,30]. This footprint of urban vacant land with potentially high detention capacity exists in many cities that are simultaneously struggling to manage excessive stormwater inputs into ageing centralized wastewater collection systems with finite capacities[26,37–39]. Though cities are already adopting strategies to incorporate urban vacant land into their planning and stormwater management strategies[15], there is limited scientific evidence as to whether urban vacant lands may provide hydrologic benefits. Much of this evidence is found only through assessment of soil properties and land cover at the parcel level, to account for services rendered citywide.

In cities where vacant lots are an expanding component of the urban landscape, we theorize that the contribution of individual vacant lots may cumulatively provide citywide benefits for stormwater management. We hypothesize that the potential RDT across vacant lots is high, partitioning a greater volume of precipitation into infiltration than runoff and therefore detaining potential excess flows into stormwater management systems. To test this hypothesis, we leveraged a unique, extensive dataset of land cover and soil physical properties across 520 vacant parcels in Buffalo, New York (NY), USA to evaluate their contribution to citywide detention capacity. Buffalo is one of many so-called shrinking cities where massive population declines have left many properties vacant or abandoned (Supplementary Fig. 1), and where citywide efforts have been devoted to reducing vacancies through demolition[2,5,8,9].

In this analysis, we sought to document the potential benefit of urban vacant parcels for stormwater management, synthesizing qualitative and quantitative data characterizing vacant lot form (land cover, topography) and function (infiltration rates) and incorporating averaged, measured parcel infiltration rates (estimated as maximum infiltration rates) and pervious lot area into the application of a simple infiltration-excess model. We apply this model and a synthesis of field observations to test for spatial structure in RDT, to determine thresholds of storm magnitude and intensity that initiate runoff from these properties, to identify classes of vacant lots that most effectively infiltrate rainfall, and to quantify changes in detention capacity provided by vacant land at citywide scales (extending a smaller dataset, $n = 520$, to ca. 2400 vacant parcels across Buffalo). As we show, vacant parcels present as well-vegetated with low slopes and infiltration rates that often exceed rates of warm-season precipitation. When extrapolated to a city scale, we estimate that the reduction of impervious surfaces, represented by former building footprints, may lead to gains in infiltration and reductions to runoff on vacant parcels; gains persist regardless of warm-season precipitation regime. Our analysis documents the evolution of vacant lands (including the effects of redevelopment) as landscape-level stormwater control measures, and—critically—quantifies the impact of these measures on hydrologic partitioning at the land surface.

## Results

**Characterizing vacant parcel infiltration rates**. Parcel-level infiltration rates ranged from 0.001 to 5.39 cm h$^{-1}$ ($n = 520$; mode: 0.001 cm h$^{-1}$) across fourteen sewersheds (Fig. 1; Supplementary Table 1). When compared across the city, parcel-level infiltration rates had weak spatial structure (Fig. 1; Supplementary Tables 2 and 3). Notably, parcel infiltration rates within all sewersheds varied by three orders of magnitude (Fig. 1). We found no evidence linking variations in demolition technique (e.g., how contractors approached the demolition process or sourced soil backfill material) with measured infiltration rates (period: 2001–2013, Supplementary Fig. 2, Supplementary Table 4).

**Dynamic infiltration responses to rainfall**. Our analysis relies on comparisons between historical rates of rainfall and measured rates of infiltration. We estimated RDT, expressed in terms of flux (length per unit time), as a function of rainfall rate, a maximum infiltration rate, and areal extent of pervious surfaces, all of which will determine the capacity of these parcels to absorb rainfall. Maximum infiltration rates were determined from field

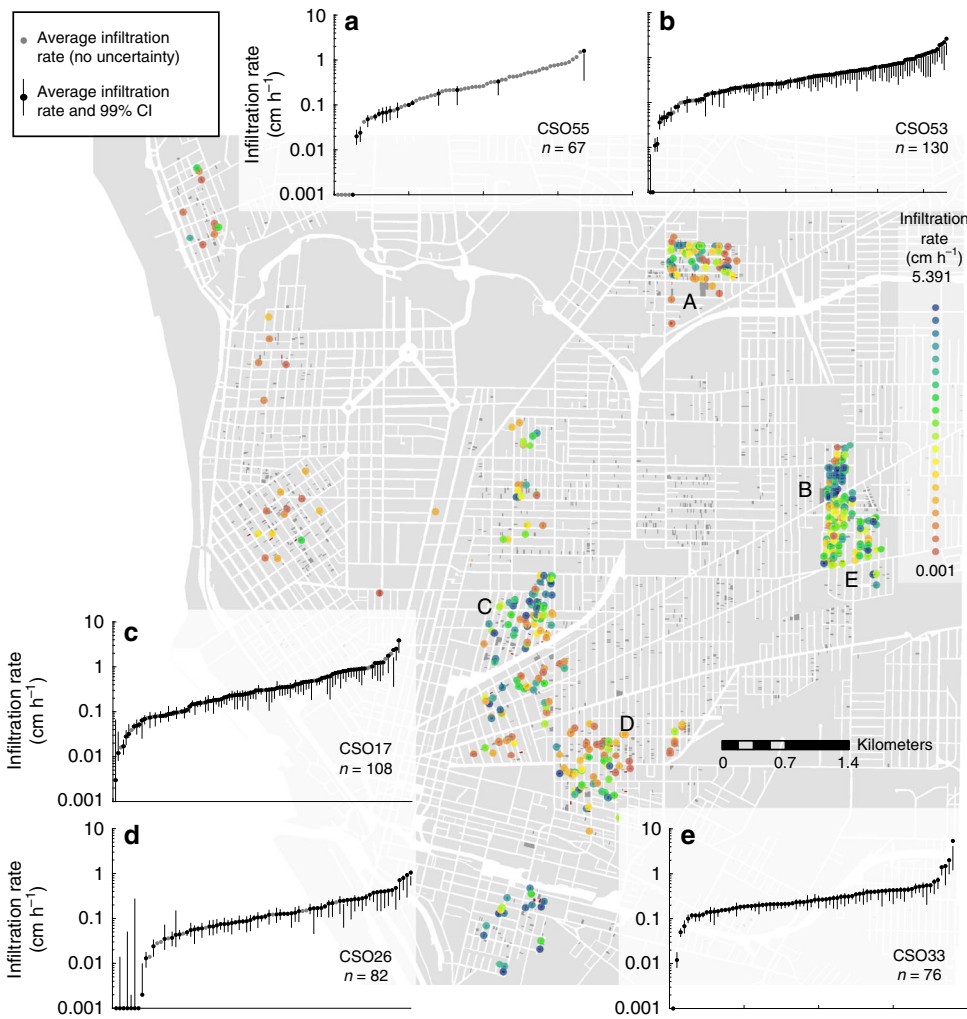

**Fig. 1 Average parcel infiltration rate (n = 520) across Buffalo, NY (USA).** Inserts display distributions of infiltration rates for five sewersheds, including **a** CSO 55 (n = 67), **b** CSO 53 (n = 130), **c** CSO 17 (n = 108), **d** CSO 26 (n = 82), and **e** CSO 33 (n = 76) with associated uncertainty estimates as 99% confidence intervals for a subset of sites (n = 426).

observations as unsaturated hydraulic conductivity (K (−2 cm)). Across the historical rainfall record for Buffalo, NY (1997–2017), we determined that 93% of all warm-season (April to October), hourly rainfall intensities were less than 0.5 cm h$^{-1}$ (Fig. 2). Based on this analysis, we found clear evidence that parcel-level detention capacity is—by comparison—high, such that infiltration rates for ~80% of vacant parcels equaled or exceeded the median historical hourly precipitation intensity (0.08 cm h$^{-1}$) for warm-season rainfall, and rates on ~60% of parcels equaled or exceeded the 75th percentile hourly precipitation intensity (0.18 cm h$^{-1}$). The most frequent rainfall events in the warm season were estimated to be entirely detained in vacant lots via soil profile-control of the infiltration process.

Based on the same 21-year, warm-season precipitation record, we approximate that the ca. 500 vacant lots could infiltrate between 15.5 (0.01 cm h$^{-1}$) and 83.9 m$^3$ (0.056 cm h$^{-1}$) for a representative one-hour storm event spanning the most frequent, low- intensity hourly rainfall rates (0.01–0.05 cm h$^{-1}$). These lots were estimated to infiltrate more than 546 m$^3$ per event for the less-frequent hourly events that exceed 1 cm rainfall depth (Fig. 2). To derive these estimates, we summarize parcel hydrologic condition using infiltration rates applied to pervious lot fractions. We partitioned rainfall using observed infiltration rates as a maximum value above which any precipitation would become direct runoff (generated by infiltration-excess) and below which any precipitation was assumed to be infiltrated into the subsurface. The resulting simulated RDTs, therefore, represent the aggregated volume of infiltrated water for precipitation events with a 1-h duration and variable intensity.

Across all parcel-averaged infiltration rates, our calculations indicate that the most frequent, low-intensity rainfall events are fully detained or generate minimal runoff for the vast majority of storms (Fig. 2). At rainfall intensities at or below 0.01 cm h$^{-1}$, we estimate vacant lands infiltrate between 91 and 94% of precipitation volumes. However, for storms with rainfall rates exceeding 0.5 cm h$^{-1}$, a threshold where rainfall rate tended to exceed infiltration rates, we expect runoff production is initiated and likely to predominate. Though this scenario favors partitioning to surface runoff over infiltration, high-intensity storms are rare in this system. For example, storms with precipitation rates exceeding 1 cm h$^{-1}$ occurred less than 2% of the time over the selected historical record. Furthermore, only 4% of parcels had minimum infiltration rates (ca. 0.01 cm h$^{-1}$). These combined circumstances suggest that estimated volumes of surface runoff are likely small compared to the overall gain in detention capacity across all parcels.

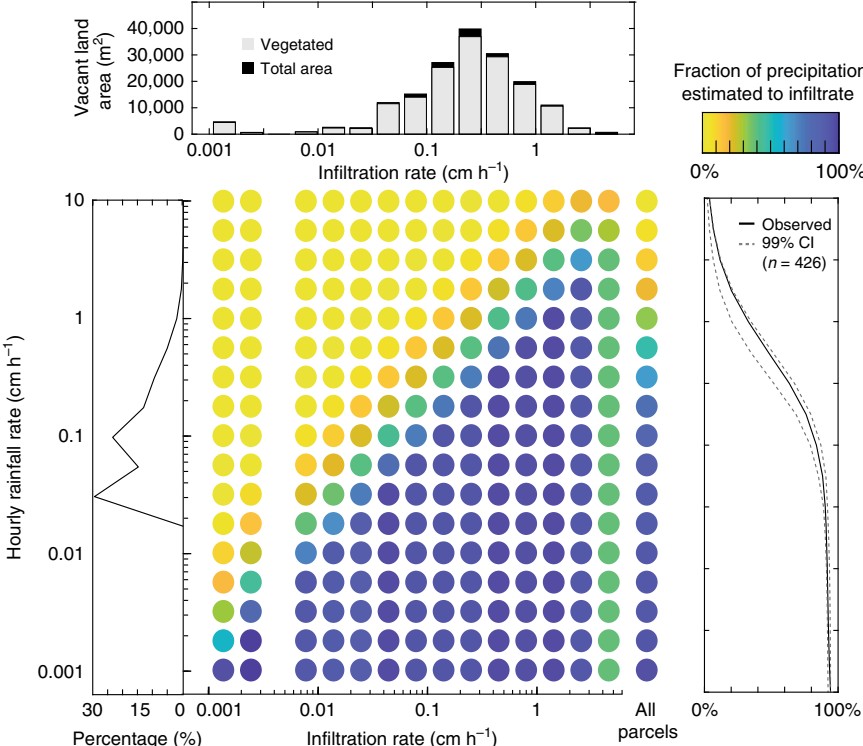

**Fig. 2 Rainfall detention capacity.** Visualized as the fraction of hourly precipitation (histogram for 21 years of warm-season precipitation) estimated to infiltrate (n = 514) as a function of infiltration rate and pervious lot area. Confidence intervals (99%) show the impact of measurement uncertainty for a subset of sites (n = 426; Supplementary Figs. 10 and 11).

**Common characteristics of vacant parcels.** Our approach to estimating the hydrologic benefits of vacant parcels is rooted in a conceptual model that integrates measurements of land cover, soil resistance, and topography. These are identified and depicted in hierarchical form to describe the average presentation of a Buffalo, NY vacant parcel (Fig. 3). While the infiltration rate determines how quickly any surface water will move into the soil, the form of vacant lots may also shape the function of these areas to suppress runoff and favor infiltration. Beyond infiltration rates alone, we found that several vacant lot characteristics likely promote rainfall detention capacities and suppress runoff, namely: the presence of vegetated land cover, minimal soil compaction over a majority of each parcel, and low parcel slopes. The characteristics that support our conceptual model also indicate that vacant lots display a common internal soil and landform structure that can act to suppress runoff production, through heightened RDT.

Despite localized imperviousness, all lots were well-vegetated with low topographic gradients (Fig. 3; Supplementary Table 5). Vacant parcels, following demolition, were initially planted with a mixture of fescue and white clover, though common species observed across vacant lots included crabgrass (*Digitaria spp.*), knotweed (*Fallopia japonica;* present on 26% of fully assessed parcels), garlic mustard (*Alliaria petiolata*; present on 26% of fully assessed parcels), and common lambsquarter (*Chenopodium album*). These ruderal plants aggressively establish and spread, and effectively contribute to increasing the vegetative cover in vacant parcels. Their success will also impart surface roughness, which is expected to limit further routing of localized ponding or runoff production.

Nearly 88% of lots drained toward the street; average slopes across all parcels were less than 1.5% (Fig. 3). These average slope and aspect observations are typical for neighborhoods sewered by curb-and gutter wastewater collection systems, where the intent is

to route surface runoff toward street inlets. We found from slope data that the demolition process did not modify the parcel slope from its built condition. We expect the gentle slope of the land surface to slow runoff velocities (compared to steeper slopes), allowing greater contact time for runoff to pond at the surface and either infiltrate or evaporate. This assumes that a greater fraction of precipitation than estimated via our simple modeling may be retained on these parcels, and not routed via the curb-gutter-street network to storm sewer inlets. The presence of canopy (e.g., via evapotranspiration) and understory (e.g., via flow resistance mechanisms) vegetation and micro-topographic depression storages may further contribute to low runoff production and therefore high detention capacities from these parcels.

While 77% of vacant parcels were entirely vegetated, a subset of parcels contained impervious surfaces left behind by an incomplete demolition process. In some cases, impervious surfaces occurred as small, disconnected patches (i.e., surrounded by pervious surface), while other parcels contained impervious surfaces nearer to sidewalks and abutting adjacent streets. Our near-surface soil penetration resistance measurements identified other demolition activity artifacts that manifested as compact, denser soil structures prominent at the street-facing front of the typical parcel (Fig. 3; Supplementary Fig. 3). These arrangements of impervious and compacted soils may expand localized coverage of connected (or effective) impervious surface, given that roadways drain to the centralized wastewater collection system. However, the vast majority of the typical lot area was vegetated with less-compact soils, which combine to favor infiltration and storage opportunities, and thereby suppresses runoff.

In urbanized areas, characterizing reference, pre-urban land use and soils can be difficult. Similar to earlier work[40] and to address this challenge, we compared common urban soil units

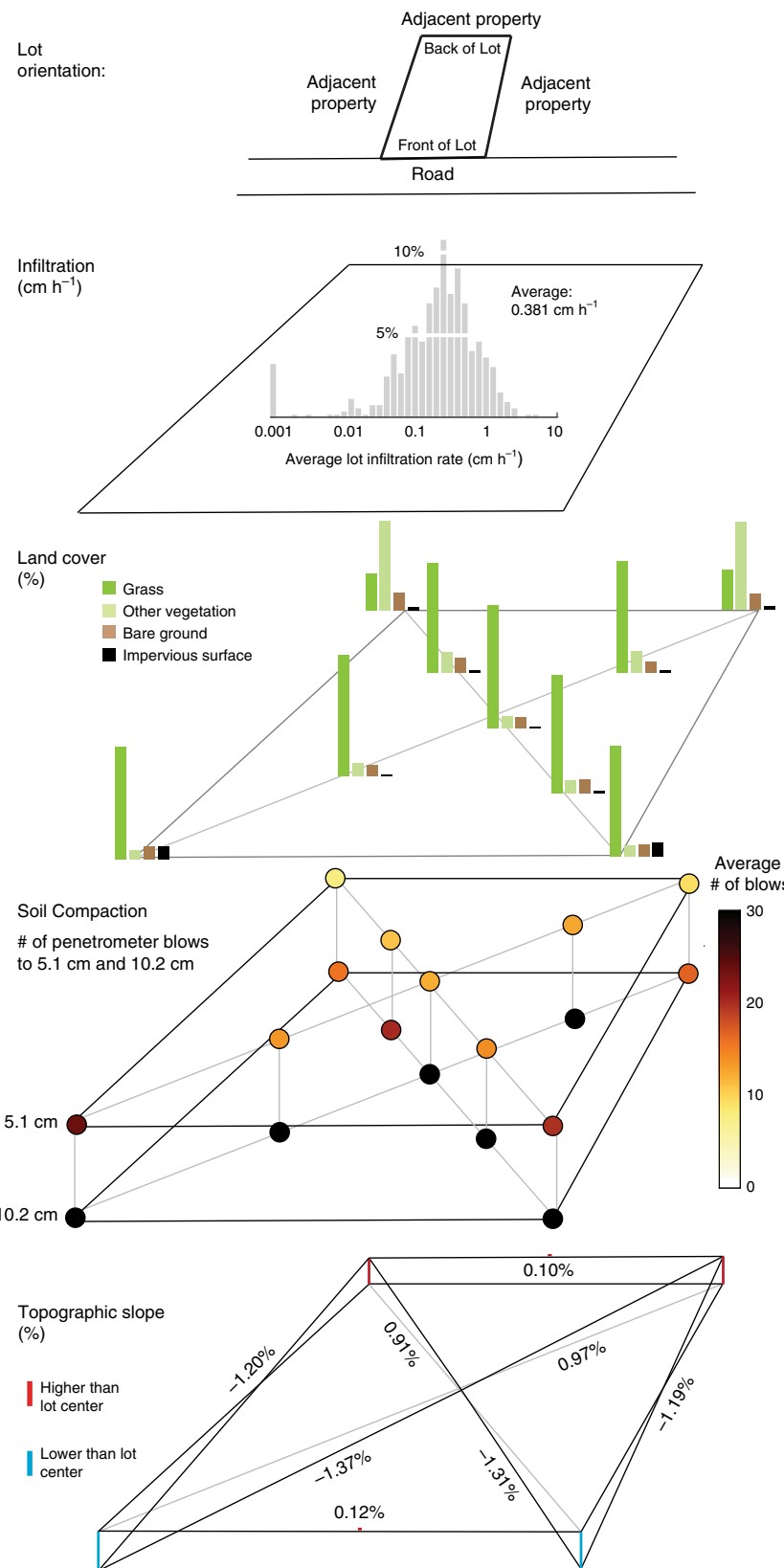

**Fig. 3 Sub-parcel characteristics summarized across vacant parcels.** These characteristics, including distributions of land cover ($n = 709$), average soil compaction ($n = 715$), and average topographic slope ($n = 713$), will interact with infiltration rate ($n = 520$) to favor infiltration versus runoff on vacant parcels. These characteristics, measured and mapped at multiple sites across each lot, reveal the conventional form and functional presentation of vacant parcels.

and their reference soils, available from the Soil Survey Geographic Database (SSURGO). This comparison benchmarks how our assessment compares with SSURGO estimates, which serve as the best information available for Buffalo, and other cities[40]. As we show, inflated infiltration rates from these reference soil units would lead to large overestimates of detention volume (Supplementary Note 1; Supplementary Figs. 4–6; Supplementary Table 6). This circumstance underscores the value of high resolution field observations to inform the critical planning stages of re-development, and ongoing efforts to manage runoff. Our approach provides a robust starting point for science-based planning and infrastructure-asset needs assessments.

**Citywide benefits of vacant parcels**. When taken together, the characteristic internal structure of assessed parcels suggests that vacant land in Buffalo is a hydrologic sink for rainfall. Our results indicate that these areas are a resource that can be leveraged as decentralized green space contributing to the prevention of runoff formation. These areas interrupt runoff routing to the urban stormwater-wastewater management system. To further confirm this, we randomly resampled our representative assessment of average parcel hydrology ($n = 520$) across a larger dataset of unassessed vacant parcels to calculate cumulative, citywide RDT (Fig. 4). This estimation accounted for ca. 2400 decentralized vacant parcels dispersed over an area of $0.85\,km^2$. Using historical imagery from 2002, our conservative estimates of former building footprints suggest that on the order of $0.29\,km^2$ of pervious area was gained across a combined 2482 lots. Accounting for redevelopment and incomplete demolition (ca. 2017), we estimate $\sim 0.0028\,km^2$ of impervious surface across these areas; this estimate may continue to grow with ongoing redevelopment.

Our analysis indicates that demolition of buildings on ca. 2400 properties (comparing pre- to post-demolition) has increased the cumulative rainfall detention rate by an average of 52% (range: from 51 to 54%) across the city (Fig. 4; Supplementary Tables 7 and 8). These numbers are impressive when considering the estimated footprint of former impervious cover is only a fraction (~34%) of the total vacant parcel area in Buffalo, NY. When accounting for redevelopment, our results still suggest a net gain in infiltrated volume between 50 and 54%. During a wetter year, we estimate that nearly $436,000\,m^3$ (range: $420,000–452,000\,m^3$) of rainfall would be infiltrated and detained on vacant parcels; for a drier year, the estimate is accordingly lower at $225,000\,m^3$ (range: $217,000–232,000\,m^3$) (Supplementary Tables 9 and 10). While volumes of infiltration and runoff varied with the magnitude of annual precipitation, precipitation-normalized gains in infiltration and reductions in runoff were relatively consistent from year-to-year (Supplementary Tables 7 and 8). These stormwater management benefits persist regardless of the precipitation regime, demonstrating that the vacant land mass provides rainfall detention benefits citywide. This landscape retention of rainfall likely translates to reducing direct runoff volume inputs to local sewer networks.

In Buffalo, the expanse of vacant land disconnects or eliminates a substantial amount of connected impervious surface (~34% of total vacant land area). Altogether, this creates a comparatively large area of hydraulically-disconnected land mass with characteristics that favor detention (Fig. 4b). Considering that our estimates are based on 2482 parcels representing only a fraction of the thousands of parcels that have been demolished across the city (Supplementary Fig. 1), our assessment represents an underestimate of the overall benefits from demolition, even when considering redevelopment.

## Discussion

The relative abundance and physical attributes of vacant land has implications for cities of all sizes, particularly those where demolition is a part of redevelopment and blight control[5,8,9,15]. The character of vacant parcels is molded by human disturbance through the introduction of soil fill, compaction from heavy machinery, and remaining relics of former land use (e.g., sidewalks, driveways). Though disturbance can modify, and at worst degrade, the net permeability of the soil profile[17,40–43], we found that almost 60% of parcels in Buffalo had infiltration rates greater than the 75th percentile of historical hourly rain intensity (Fig. 2). Only a fraction (~2%) of surveyed parcels were observed to have impermeable surfaces over 50% or more of the parcel area. Urban vacant parcels in Buffalo displayed a common structure of land cover, compaction, and geometry that reflect a complex history and evolution. Nearly all vacant parcels were well-vegetated and gently sloped. Highly-compact soils were limited to locations close to roadways where demolition activities and traffic are concentrated (Fig. 3). Estimates of cumulative, citywide RDT suggest that vacant land expansion across Buffalo may result in undocumented benefits and an untapped resource for stormwater management (Fig. 4).

Our study is the first, to our knowledge, to quantify citywide vacant parcel infiltration rates and to translate these rates into rainfall detention capacities at storm and seasonal scales. Though we found that infiltration rates across the city were highly variable (Fig. 1), these rates generally exceeded rainfall rates across the majority of vacant parcels and were estimated to infiltrate and detain sizable fractions and volumes of rainfall for both hourly and (Fig. 2) and annual (Fig. 4) periods. We estimate that this detention capacity is greatest for the most frequent, small rainfall events ($< 0.5\,cm\,h^{-1}$).

The unique interspersion of pervious vacant parcels with impervious surfaces creates a patchwork of connections—and disconnections among impervious surfaces—that contribute to citywide hydrologic benefits. Our observations that greened vacant parcels may favor infiltration and reduce runoff place these urban green spaces in the context of a larger body of literature, which documents runoff reduction via green infrastructure[44–46], lawns[47–49], and street trees[49,50] across cities. Though our calculated detention volumes consider gains of pervious surface and detention capacity within the footprint of each lot, we did not account for other impervious structures (e.g., driveways, curb cuts, roads) in our pre-demolition estimates of impervious cover. These pre-demolition impervious areas likely represent additional gains in pervious cover through the demolition process, particularly because they are designed to route water to streets and other impervious surfaces outside of each parcel. Therefore, we postulate that vacant parcels may yield additional stormwater benefits beyond those we estimate in this analysis and that our estimates are conservative.

Though we focus on the partitioning of rainfall between infiltration and surface runoff, detention capacity may be indirectly altered through deep drainage and evapotranspiration[25,51]. As vegetation intercepts rainfall that would otherwise reach impervious areas[21], the presence of vegetation on vacant parcels may amplify the RDT of these areas. Vegetation may also increase warm-season soil moisture loss through evapotranspiration processes. This suggests that detention capacity may be underestimated in our analysis.

The urban subsurface will also shape the efficacy of vacant parcels to serve as stormwater resources[17,40,51]. However, the impact of subsurface characteristics and below-ground hydrologic processes (e.g., deep drainage) on detention capacity represents a major source of uncertainty because subsurface hydrologic data are limited in most urban areas (Supplementary Note 2). For

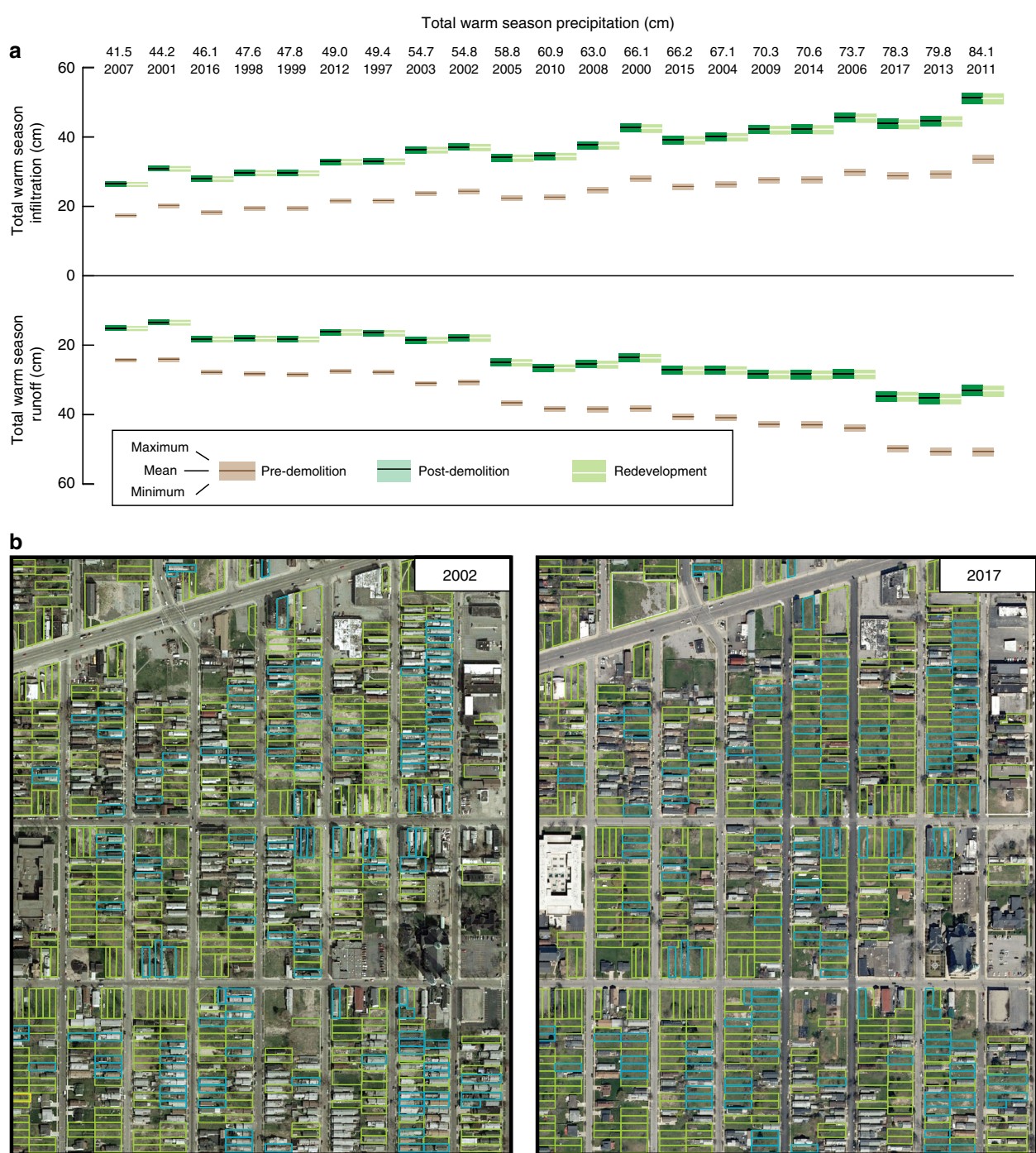

**Fig. 4 Interactions between pervious area and precipitation partitioning between infiltration and runoff for 2482 lots.** Values are presented as **a** cumulative depths (normalized by parcel area) for three end-member states (pre-demolition, post-demolition, redeveloped) that have **b** introduced a patchwork landscape of green space across the city. Ranges show minimum and maximum values obtained by resampling infiltration observations 1000 times.

example, we did not directly incorporate vacant lot subsurface architecture and associated macroporous spaces into our analysis, although these characteristics can influence the dynamic distribution of soil moisture in the soil profile[24]. We also assume free drainage as a lower boundary condition. This assumption will be violated in areas where soil moisture is concentrated or forms a water table (e.g., localized groundwater mounding), predisposing the landscape to generate runoff by saturation-excess[52], or form variable source areas[53]. Though the majority of Buffalo is located

in lacustrine silt and clay surficial geology, shallow water tables were historically observed in only a few locations, and water table depths across the majority of the city were typically deep (Supplementary Figs. 7, 8). As little is known about the urban subsurface in Buffalo and other urban areas, we recommend expanded, standardized field assessments to catalog and categorize vacant parcels, including their subsurface hydrology[17]. Such assessments will provide training datasets that improve the representativeness and generalizability of simulation models that

scale vacant lot detention capacities to cities. The assessments would also refine our understanding of locations in the urban landscape that contribute to surface runoff generation[24]—and how these source areas spatially vary and temporally interact with changes in local precipitation patterns to alter the mechanisms, frequency, and magnitude of urban runoff generation.

While vacant lots may achieve stormwater benefits, retaining precipitation and runoff in these areas spells unclear impacts for disturbing or introducing contamination in the subsurface. As with green infrastructure[54], a common perception associated with stormwater infiltration on vacant lots is that there may be increased risk from infiltrating contaminated water or mobilizing contaminant transport in the subsurface. While many vacant lands are formerly residential, others may have been used for commercial or industrial purpoposes[9,16] (as in Buffalo, e.g., Supplementary Table 11; Supplementary Note 3). Likewise, changes to parcel microtopography generated by demolition may introduce new pathways of surface water routing from roads to vacant lots, potentially carrying and infiltrating a host of contaminants carried within stormwater[55]. More work is needed to understand interactions amongst surface and subsurface water quality and contamination within vacant parcels, to ensure that these landscapes do not pose a threat for future residents or recreators on these lands.

Research is also needed to further disentangle the complex interactions between human-derived and natural landscape dynamics that regulate how vacant lot hydrologic benefits are rendered[5]. The evolution of vacant lands must be balanced with social equity, maintenance of civic services, and stabilization of real estate values at the neighborhood level[7,56]. As vacant lands are likely to be areas of redevelopment[8] (Fig. 4), we encourage all cities to ensure that any redevelopment efforts on vacant lands are equitably distributed throughout a given city. This underscores the importance of examining the life cycle of vacant lots (and surrounding areas) that accounts for social and economic change alongside the perceptions and perspectives of local stakeholders. This work may also include examination of how local, regional, and national economic conditions and the response of city governance can in part shape the production of vacant land.

Vacant land may be a particularly important stormwater management resource in urban settings such as Buffalo, where lower overall rainfall depths and intensities favor infiltration over runoff. Yet, we advise that any concerted planning or demolition effort to treat vacant lands as a stormwater management tool should ultimately consider both macro- and micro-features of the landscape. Each city will have a unique combination of rainfall pattern, soil properties and associated heterogeneity, surficial geology, and vegetation that must be considered before any attribution of stormwater detention benefits is made to these spaces. Likewise, as these properties and context vary extensively from parcel to parcel, obtaining more detailed information is key, and one of the foundational drivers for making these assessments. This may be particularly true in terms of how soils are layered in the subsurface (e.g., Supplementary Fig. 5). A comprehensive surface-subsurface assessment will help to identify parcels with the highest storage capacity, and ensure that those with the lowest capacity are relegated to other land uses, such as redevelopment. In the case where there are locally high water tables, the possibility of their regulation of subsurface storage potential is important, especially in terms of assuming runoff formation via infiltration- than saturation-excess mechanisms.

Many cities are alive with green spaces that are valued for numerous ecosystem services[57,58]. In this way, neighborhoods may consider a portfolio approach to stormwater management, interspersing residential lawns, green infrastructure, and vacant landscapes to create greater connectivity of green spaces, likely with numerous benefits for city inhabitants when distributed equitably. Therefore, future research may also consider the role of residential lawns alongside vacant lands. Though the stormwater capacities of residential lawns have been widely investigated[47,59,60], we currently have little to no baseline for comparison regarding how infiltration rates on vacant lots compare with nearby residential lawns. Field studies that have linked construction date (as a proxy for last date of soil disturbance) to infiltration rates on residential lawns suggest that if properly maintained, many residential green spaces are likely to provide additional capacity for stormwater volume, beyond those attributed to vacant lands[48,59,60]. Many cities already take a portfolio approach to planning and installing different sizes and types of GI. Adding vacant lands and even residential lawns to these planning efforts may be a natural extension that offers greater flexibility[15].

We conclude that broadening the footprint of pervious cover from urban vacant land is a tangible stormwater resource with the potential to retain large water volumes on a seasonal and annual basis. Vacant lands provide the greatest retention benefits for low-intensity rainfall events (Fig. 4); these conditions promote infiltration and prevent formation of surface runoff. However, future precipitation projections suggest the ascendance of storms with both higher intensity and larger overall rainfall depth in the Northeastern U.S.[61,62]. While our work intimates that vacant lands will be pushed into runoff production under these storm patterns, they will still provide a net benefit by detaining precipitation up to a threshold rainfall depth, and reducing and delaying the delivery of runoff to sewer systems. The resilience of vacant parcels to more strenuous rainfall forcing may further be enhanced by proactively interspersing vacant land with retrofit-designed GI throughout the urban landscape, which may scale to citywide cumulative effects.

Though all cityscapes evolve differently, as do their soils, we may expect equivalent hydrologic benefits in cities experiencing similar cycles of socio-ecological disturbances that lead to the expansion of vacant land mass. Our study underscores that the partitioning of precipitation between hydrologic pathways in urban areas is intricately linked to human activity, land cover, and land use. Future investigations into these pathways will likely bolster and expand our present findings that urban vacant lands can render several tangible biophysical and socio-economic services[15,17,18,23]. Most importantly, these areas represent a global opportunity to integrate, transform, and re-invest in the urban landscape.

## Methods

**Vacant parcel assessment.** To assess the thousands of demolitions that have occurred across Buffalo, NY (USA; Supplementary Fig. 1), an Urban Vacant Land Assessment protocol was developed and applied to vacant parcels across the city[63,64]. This protocol is a per lot assessment of general site information, parcel size, topography, vegetation, wastes, soil type, and hydrologic and land cover surveys of pervious and impervious surfaces (Supplementary Fig. 9; Supplementary Note 4). As part of the extended protocol, measurements of surface infiltration, soil type, land cover, and topography were made across each lot across 718 vacant parcels (Supplementary Table 12) within 15 sewersheds (Supplementary Table 1). Topography and land cover were assessed at nine station points across each lot (Supplementary Fig. 9). The percentage of each lot covered by vegetation, taken to represent the fraction of pervious surface, was assessed on a scale from 0 to 100% with 25% increments (Supplementary Table 5). Soil compaction was assessed using a single mass penetrometer at a minimum of four locations per lot, with the assessor recording the number of blows to 5.1 cm (one increment), 10.2 cm (two increments), and 15.2 cm (three increments) at selected stations or until refusal. Soil texture and infiltration rates were assessed at two locations per lot, the lot center (capturing the area impacted by demolition) and lot rear (outside the demolition envelope), and averaged for 520 vacant parcels. Soil texture was assessed by feel[65] (Supplementary Table 13). Maximum infiltration rates were calculated from measured volumes of water infiltrated through time recorded with a Mini Disc Infiltrometer (© METER Group Inc), which enables a rapid approach

to determine infiltration rates as unsaturated hydraulic conductivity[66]. Maximum infiltration rates were estimated as unsaturated hydraulic conductivity[67] by fitting rates of cumulative infiltration as water volume through time using a simple infiltration model incorporating van Genuchten parameters for a given soil type[68], the radius of the infiltrometer, and the suction at the disk surface (set during each field experiment to -2 cm). Further details are included in Supplementary Note 4. Like all models, this simple approximation has limits, and generates negative rates for very slow infiltration rates. As this represents a physical impossibility, negative rates were converted to the smallest value measured in the dataset, 0.001 cm h$^{-1}$. Assessments were paired with existing information on the date of demolition, to determine whether infiltration rates were related to demolition type (Supplementary Fig. 2), and parcel geometry, to extract land surface slope (Fig. 3). Infiltration rates were subject to an assessment of uncertainty, with reported confidence intervals (99%) shown in Supplementary Figs. 10 and 11.

**Pre-demolition and redevelopment states.** Aerial imagery across Buffalo was leveraged to hand-delineate former building footprints on 2482 vacant parcels for two time points (2002, 2017). Orthoimagery from 2002 was used to establish a pre-demolition estimate of former impervious cover, assumed to be an underestimate given that many parcels also included sidewalks, driveways, disconnected garages, and sheds. Sites demolished prior to 2002 were excluded from analysis. Imagery from 2017 was used to establish a more recent estimate of building footprints (assumed to be a good approximation for impervious cover) on vacant properties, accounting for incomplete demolition or redevelopment.

**Estimation of rainfall detention capacity.** To estimate volumes of RDT across each vacant lot, we applied a simple model of infiltration, taking average parcel infiltration as the maximum rate ($I_{max}$) that water can be infiltrated over the percentage of pervious area (V) per parcel. In this model, we assume unvegetated surfaces (impervious surface, bare ground) cannot contribute to RDT and will contribute to the volume of overland flow (OF). When parcel infiltration exceeds hourly precipitation intensity (P), RDT and OF were calculated according to:

$$\text{RDT} = P \cdot (V/100) \cdot A \qquad (1)$$

$$\text{OF} = P \cdot (1 - V/100) \cdot A \qquad (2)$$

where A is parcel area. When hourly precipitation intensity exceeded parcel infiltration, RDT and OF were calculated according to:

$$\text{RDT} = I_{max} \cdot (V/100) \cdot A \qquad (3)$$

$$\text{OF} = (P - I_{max}) \cdot A \cdot (V/100) + P \cdot (1 - V/100) \cdot A \qquad (4)$$

where Equation 4 represents OF generated from pervious areas (when $P > I_{max}$) and from impervious areas (when $P > 0$). This model makes several assumptions, which are described in Supplementary Note 2 and tested on the basis of available historical data and vacant parcel observations (Supplementary Figs. 7, 8, 12, 13).

Hourly design storms used to estimate RDT and distributions of P (Fig. 2) were compiled from hourly timeseries of warm-season precipitation (April–October) for 1997 to 2017 from Buffalo Niagara International Airport[69]. Prior to analysis, hourly precipitation data were compared with daily data from the National Climatic Data Center to ensure observations were comparable (Supplementary Table 14). RDT estimates are contextualized with an uncertainty assessment of 426 properties (82% of infiltration dataset), obtained by propagating an estimate of measurement error through water levels to estimates of infiltration (Supplementary Figs. 10 and 11). These uncertainty bounds were used to provide an envelope of uncertainty on the fraction of precipitation estimated to be partitioned between runoff and infiltration for hourly design storms (Fig. 2).

**Scaling estimates citywide with historical precipitation.** We extended this simple model to assess how RDT would vary under a pre-demolition scenario (ca. 2002), a post-demolition scenario, and a redevelopment scenario (ca. 2017) in response to variable warm-season precipitation regimes. For the pre-demolition state and redevelopment state, V was assumed equal to A minus the area of each building footprint. Building footprints were assumed to be entirely impervious and to contribute directly to runoff. For some sites, this may overestimate runoff and underestimate infiltration, as not all roofs will route water directly to the road network and eventually stormwater sewers. However, given that most parcels have been and continue to be built near the front of each lot, abutting sidewalks and driveways, and parcels generally slope towards the street, we believe this simplifying assumption represents a reasonable starting point for estimating surface partitioning of precipitation in the presence of structures. The post-demolition state assumes the entire lot is pervious, and represents a hypothetical upper bound on estimates of infiltration and runoff across this dataset.

Our analysis for 520 properties was extended to 2482 sites with known parcel areas and delineated building footprints. We randomly resampled the distribution of infiltration rates from 520 properties to these 2482 properties 1000 times, and subsequently calculated RDT across warm-season precipitation from a twenty-one-year record. We assumed a maximum rate of infiltration at the surface for each timestep. Figure 4 displays RDT across a concentrated period of blight control and

initial redevelopment, demonstrating the cumulative impact of demolition (Fig. 4). Results for annual precipitation regimes (1997–2017) are tabulated in Supplementary Tables 7–10.

## Data availability

Raw data and derived data shown in figures are available at the Environmental Protection Agency Science Hub repository [https://doi.org/10.23719/1503451][70]. Exact lot location information is proprietary information of the Buffalo Sewer Authority. Our analysis has also leveraged publicly available property inventory (2018) provided by Erie County (http://gis.ny.gov/parcels/) and historical orthoimagery from the Discover GIS Data NY site: http://www.orthos.dhses.ny.gov/.

## Code availability

No specialized custom code or mathematical algorithms were used. All equations leveraged in the paper are described in the Methods section.

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

## Acknowledgements

The authors would like to acknowledge several contributors to the Urban Vacant Land Assessment Protocol. Foremost, protocol development and application to sites across Buffalo would not have been possible without the Buffalo Sewer Authority as well as numerous undergraduate and graduate students who spent summers working to apply the Urban Vacant Land Assessment Protocol to sites across Buffalo. This work was supported by the US Environmental Protection Agency (USEPA) Technical Assistance Programme [contract number EP-C-11-009]. This work was also supported by the National Science Foundation Cooperative Agreemtn 1444755, the Urban Resilience to Extremes Sustainability Research Network. This paper has been reviewed in accordance with the U.S. Environmental Protection Agency's peer and administrative review policies and approved for publication. The authors would like to thank Chris Knightes (USEPA) for helpful comments on an earlier draft of the paper. Mention of trade names or commercial products does not constitute endorsement or recommendation for use. Statements in this publication reflect the authors' professional views and opinions and should not be construed to represent any determination or policy of the U.S. Environmental Protection Agency.

## Author contributions

S.B. and W.S. conceived of and designed the data collection scheme and led collection of the data. C.K., H.G., and W.S. performed analysis of the data and wrote the paper, with input from S.B.

## Competing interests

The authors declare no competing interests.
