## [Peer Review File · Nature Communications]

Reviewers' comments:

Reviewer #1 (Remarks to the Author):

This is a very interesting paper that evaluates the utility of an urban problem (vacant, leveled parcels) as an urban resource. It is well written, and addresses an important societal problem in a novel way. The evaluation of the infiltration capacity of the numerous vacant parcels in Buffalo, a city with a historically challenged economy following manufacturing, shipping and significant population decline, provides important insights into unintended, and potentially valuable cumulative ecosystem services. The authors provide information on an expanding number of vacant lots in Buffalo up to ~2014, make detailed measurements of infiltration capacity on a sample of these, and estimate the cumulative volume of stormwater reduction potential under different rainfall intensities, relative to the warm season precipitation frequency distribution. Interestingly, Buffalo appears to now be experiencing an economic revitalization, with a recovering housing market due to the overall national recovery, and specific investment in Buffalo by NY State and private groups. This may reduce the number and area of vacant lands, as experienced in the downtown core that has seen significant construction.

I have a set of suggestions that would strengthen this paper:

1. It is not clear how much of the vacant lot infiltration capacity is reducing potential stormwater production as there were no estimates of occupied lot infiltration capacity. While much of Buffalo has narrow residential lots with high impervious cover, there are also lawns, shrubs and other vegetated pervious area. The paper does not provide estimates of what the pre-demolition land cover was. No infiltration capacity measurements were made on occupied lots. Therefore it is difficult to estimate how significant the potential reductions are. Lawns can have high infiltration rates if properly managed, and we do not have estimates of impervious/pervious cover on occupied lots.
2. An estimate of the volume of precipitation infiltration on sampled vacant lots is provided, then extended to all vacant lots. The potential infiltration is given ranging from ~600,000 to 1,000,000m³ in dry and wet years, respectively. I don't know if this is a big or little number relative to the area of the sampled vacant properties, and certainly the area of the City of Buffalo. If it were provided as an equivalent depth by area normalization it would be more informative. However, it should be quantified as the amount of additional infiltration relative to intact lots. The latter step would require measurement of impervious and pervious surface cover preceding demolition, with reasonable estimates of infiltration capacity on pervious land.
3. Buffalo is built largely on lake bed, with mixtures of sands, clays and some till. At least some of the clays are thixotropic. Relatively flat topography and continental climates can yield seasonally high groundwater levels which can reduce infiltration capacity. The model for infiltration capacity assumes free drainage as a lower boundary condition. This may result in overestimation of effective infiltration by underestimation of saturated areas.
4. Are parcels entirely residential or do they include former industrial lands? Brown fields and contaminated soils occur through different parts of Buffalo, including areas where vacant lands were sampled. How much of a potential groundwater contamination do the expanded vacant land pose?

Reviewer #2 (Remarks to the Author):

This paper estimates the impact of vacant lots in Buffalo on infiltration rates. By doing so the paper quantifies an important potential benefit of these areas for flood risk and potentially groundwater recharge in this city.

The paper could be a useful starting point for evaluating the ecosystem service contribution of vacant lots in humid environments. Using these results to support the parameterization of a hydrologic model of runoff generation (and other impacts of urbanization) would be a valuable extension of the work.

My main concern with this work is the assumption that infiltrated water necessarily contributes to detention capacity. If infiltrated water follows preferential flowpaths to surface or even soil moisture in down gradient areas, the effective detention may be quite minimal at hill slope to watershed scales. Saturation from below may also be an issue and this was not addressed in the paper. In other words, this study presents convincing evidence that infiltration rates are high in vacant lots - but does not necessarily show that these high infiltration rates provides substantial hydrologic services. Further analysis is needed before concluding that improved infiltration leads to meaningful increases in "detention capacity" that reduce runoff at a city scale.

Another limitation is the lack of comparison with alternative land covers. Comparison with other types of lots - low density housing etc. is also needed before "additional infiltration" can be quantified. It is not clear in the study what the 'null' state is - what would be the infiltration of a "developed" lot in the neighborhoods with the vacant lots.

It may also be important to consider that there may be water quality issues associated with water moving through potentially contaminated soils in these vacant areas. These effects may indeed be small - these types of additional issues should at least be acknowledged in the discussion

There are some interesting results - related to characterizing vacant parcels - In particular the finding that most are "well-vegetated, gently sloped" - and the finding that exceptions tend to be "near roadways where demolition activities and traffic area concentrated" - A key question, however, is how generalizable are these finding to other cities . A more thoughtful assessment of this generalizability is needed. For example, on line 308 authors note that vacant land may be particularly important in areas where rainfall depths/intensities are lower than in Buffalo - but these regions may also have substantial differences in soil and vegetation which may change the infiltration rates of vacant lots!.

Finally soil characteristics are notoriously variable at fine spatial scales. Some assessment of variation in infiltration (as measured with the infiltrometer) using multiple samples for a few sites would provide error bounds on the mean parcel estimates. Estimates of uncertainty would strengthen this work. Are two sample per lot sufficient?

line 103-104- "different ranges of infiltration rates were co-located" - this is too vague to be meaningful

line 107 - the sentence structure implies that the lack of a relationship between demolition type and infiltration is somehow tied to the increase in demolition. - I don't think this is what is intended

line 157 Some statistics on these relationships (that support the conceptual model) would be lend credibility here (e.g what is the relationship between vegetation cover and estimated infiltration) - Given the data some additional information on types of vegetation cover might be useful to analyze (or at least describe what types of vegetation are typically found in vacant lots)

Reviewer #3 (Remarks to the Author):

The authors combine an impressive field dataset and simple hydrological models to demonstrate the potential of vacant land to contribute to green infrastructure. The work represents among the most thorough assessments of vacant lots I have seen to date, and the manuscript is well researched and clearly presented. It will make a useful contribution to the urban ecology and hydrology literature. I have only very minor suggestions.

Abstract:

The abstract would benefit from more clarity re: the relationship of the present findings to demolition techniques and regulations. Specifically:

Line 28: "Despite variation in demolition technique". This line is unclear. Wouldn't one expect

infiltration to span different rates with variation in demolition technique? Clarify whether infiltration is/isn't correlated with variation in demolition technique here.

Line 32: Unclear what the role of regulation/demolition process is, here. In what way should it be regulated? What effect would this have?

Introduction:

Overall, the introduction is clear and well written. Given the structure of the journal, the introduction would be improved by a minor increase in the methodological details provided (e.g. lines 90-93) so that the reader can better interpret the results in the following section. While this information is clearly presented in the supplemental information, one or two sentences within the introduction clarifying, e.g., which soil properties were measured and whether/how field methods were used would improve the flow of the manuscript.

Response to Reviews:

We thank three anonymous reviewers for their assessment of our manuscript, which has been expanded and revised to comprehensively address their collective comments. All reviewers were consistent that the analysis we have performed is a unique addition to the literature. All reviewers raised important points for clarifying analyses, assumptions, and the extent to which our results can be extended beyond one particular city.

By our assessment, the main thrust of all comments was the need for an assessment of a landscape-level “null state” or reference, pre-urban state as a basis for comparison with the vacant, post-urban landscape. In response, we carried out an extensive analysis to contextualize our quantitative estimate of the impact of vacant lots on detention capacity. Specifically, we performed additional analyses for the manuscript, including: (1) an assessment of reference, pre-urban soil properties as a proxy for pervious surface hydraulics. These data were accessed from the best-known sources in the USDA SSURGO-NCSS dataset (Supporting Information, Section S13); and (2) an assessment of landscapes formerly under impervious cover. Impervious area extent was estimated via hand delineation of prior building footprints on the present vacant lots, drawn from historical aerial imagery.

Based on the remainder of the valuable suggestions across these three reviews, we made substantial revisions to the text throughout, expanded our Supporting Information section, reframed all of our figures based on comprehensive reviewer feedback, and integrated our expanded analyses of a reference, pre-urban dataset.

We have addressed each of the comments below, noting locations of changes in the manuscript.

Reviewer Comments and Responses:

Reviewer #1 (Remarks to the Author):

This is a very interesting paper that evaluates the utility of an urban problem (vacant, leveled parcels) as an urban resource. It is well written, and addresses an important societal problem in a novel way. The evaluation of the infiltration capacity of the numerous vacant parcels in Buffalo, a city with a historically challenged economy following manufacturing, shipping and significant population decline, provides important incite into unintended, and potentially valuable cumulative ecosystem services. The authors provide information on an expanding number of vacant lots in Buffalo up to ~2014, make detailed measurements of infiltration capacity on a sample of these, and estimate the cumulative volume of stormwater reduction potential under different rainfall intensities, relative to the warm season precipitation frequency distribution.

Interestingly, Buffalo appears to now be experiencing an economic revitalization, with a recovering housing market due to the overall national recovery, and specific investment in Buffalo by NY State and private groups. This may reduce the number and area of vacant lands,

as experienced in the downtown core that has seen significant construction.

Response: That is an important point to note. In response to this comment (and that by Reviewer 2), we developed an additional analysis that demonstrates the value of vacant lots by comparing “pre-demolition” (ca. 2002), “post-demolition”, and “redevelopment” (ca. 2017) states across Buffalo. We used high resolution aerial imagery to delineate building footprints on vacant lots across the city for each time period.

For our pre-demolition analysis, we created a hand-delineated dataset of building footprints (n = 2482) based on orthoimagery circa 2002. We used imagery from 2017 to delineate a more recent estimate of building footprints (assumed to be a good approximation for impervious cover) on vacant properties, accounting for incomplete demolition or redevelopment (‘redevelopment state’). Our post-demolition analysis assumes 100% perviousness on all vacant lots. While some lots were subject to incomplete demolition or redevelopment, this post-demolition analysis represents an endmember assessment for comparison with estimated volumes of infiltration and runoff in the presence of former (‘pre-demolition’) and more recent (‘redevelopment’) imperviousness estimates. Our analysis suggests that the amount of impervious surface removed from the City of Buffalo is underestimated. Most homes in Buffalo have a detached car garage and may have driveways or other impervious surfaces leading up to the property.

We applied our parsimonious infiltration model to estimate surface water partitioning (precipitation, infiltration, runoff) for pre-demolition, post-demolition, and redevelopment states, assuming the presence of structures generates and routes runoff from each parcel. This intensive investigation of both the “pre-demolition” state and “redevelopment” state of this landscape is now encapsulated in Figure 4.

These new analyses called for additional text in the Methods section, and presentation of our findings (in the Results section) and discussion of their implications (in the Discussion section). Specifically, we now demonstrate that while volumes of infiltration and runoff change annually (Table S10 and S11), the fractional gain in infiltration, and proportional reduction in runoff are largely consistent from year to year (Table S12 and S13). We also confirm that water detention on vacant lots is substantially higher than detention on pre-demolition landscapes and considering the impacts of redevelopment.

Additional information pertinent to this comment is also included in the response to the next reviewer comment.

I have a set of suggestions that would strengthen this paper:

1. It is not clear how much of the vacant lot infiltration capacity is reducing potential stormwater production as there were no estimates of occupied lot infiltration capacity. While much of Buffalo has narrow residential lots with high impervious cover, there are also lawns, shrubs and other vegetated pervious area. The paper does not provide estimates of what the pre-demolition land cover was. No infiltration capacity measurements were made on occupied lots. Therefore it is difficult to estimate how significant the potential reductions are. Lawns can have high infiltration rates if properly managed, and we do not have estimates of impervious/pervious cover on occupied lots.

Response: As we discuss in the previous response, we now provide a comparison (between pre-demolition, post-demolition [maximum vacant lot distribution and perviousness], and redevelopment states) that demonstrates the value of vacant lots for increasing surface water detention capacities. We used high resolution aerial imagery to delineate building footprints on vacant lots across the city. Using this imagery, we were able to delineate building footprints on 2482 lots where homes were demolished. Our analysis suggests that the amount of impervious surface removed from the City of Buffalo is underestimated, and that rainfall detention is highest across Buffalo in the post-demolition state. This intensive investigation of both the “pre-demolition” state and “redevelopment” state of this landscapes are now encapsulated in Figure 4.

Additional text to this effect was added on page 14 lines 246 – 259:

“Our analysis indicates that demolition of ca. 2400 properties has increased the cumulative rainfall detention rate by an average of 53% (range: from 42 to 66%) across the city (Fig 4; Table S12). These numbers are impressive when considering this estimated footprint of former impervious cover is only a fraction (~34%) of the total vacant parcel area. When accounting for redevelopment, our results still suggest a net gain in infiltrated volume between 41 and 66%. During a wetter year, we estimate that nearly 434,000 m³ (range: 420,000 – 451,000 m³) of rainfall would be infiltrated and detained on vacant parcels; for a drier year, the estimate is accordingly lower at 224,000 m³ detention (range: 217,000 – 231,000 m³). While volumes of infiltration and runoff varied with the magnitude of annual precipitation, gains in infiltrated volume and reductions in runoff were relatively consistent from year-to-year (Table S12, S13). These stormwater management benefits persist regardless of the precipitation regime, demonstrating that the vacant land mass provides rainfall detention benefits city-wide. This landscape retention of rainfall likely directly translates to reducing direct runoff volume inputs to local sewer networks.”

Fig. 4: Using 21 years with distinct precipitation regimes, we (a) estimated rainfall retention across vacant land for 2482 vacant lots across Buffalo. This is presented as a cumulative volume for a pre-demolition state, taking into account the footprint of buildings on each vacant parcel, a post-demolition state, assuming the vacant lot is entirely pervious, and for a redeveloped state, as some lots either experienced incomplete demolition or have been redeveloped with new structures. Measurements from 520 lots were extrapolated to 2482 vacant lots, and resampled 1000 times per scenario to create an envelope of estimated infiltrated volume. This demolition has introduced (b) a patchwork landscape of green space across the city.

We recognize that measurements on residential lawns strengthen this analysis and provide a very different baseline for comparison. To this end, we have emphasized this area for future work in the revised manuscript on page 20-21 lines 393-404:

“Many cities are alive with green spaces that are valued for numerous ecosystem services^{49, 51-53}. In this way, neighborhoods may consider a portfolio approach to stormwater management, interspersing residential lawns, green infrastructure, and vacant landscapes to create greater connectivity of green spaces, likely with numerous benefits for city inhabitants when distributed equitably. While not considered in our analysis, future work may also consider the role of residential lawns alongside vacant lands. We currently have little to no baseline for comparison regarding how infiltration rates on vacant lots compare with nearby residential lawns. Yet, if these residential green spaces are maintained properly, these areas are likely to provide additional capacity for stormwater volume, beyond those attributed to vacant lands. Many cities already take a ‘portfolio’ approach to planning and installing different sizes and types of GI. Adding vacant lands and even residential lawns to these planning efforts may be a natural extension of these efforts and offer greater flexibility¹⁵.”

2. An estimate of the volume of precipitation infiltration on sampled vacant lots is provided, then extended to all vacant lots. The potential infiltration is given ranging from ~600,000 to 1,000,000m³ in dry and wet years, respectively. I don't know if this is a big or little number relative to the area of the sampled vacant properties, and certainly the area of the City of Buffalo. If it were provided as an equivalent depth by area normalization it would be more informative. However, it should be quantified as the amount of additional infiltration relative to intact lots. The latter step would require measurement of impervious and pervious surface cover preceding demolition, with reasonable estimates of infiltration capacity on pervious land.

Response: We have updated the assessment shown in Figure 4 to provide a baseline for comparison. Our new results now include an estimate and uncertainty bounds regarding how much water is infiltrated across vacant lots versus a “pre-development” state and a “redevelopment” state.

Based on this feedback, we have also improved Figure 2 by referencing a normalized value instead of overall volume.

Fig. 2: Rainfall detention capacity visualized as the fraction of precipitation estimated to be infiltrated for surveyed parcels ($n = 514$) as a function of infiltration rate and hourly rainfall rate. Fractional infiltrated volume relative to precipitation volume are contextualized by distributions of measured infiltration rates and hourly rainfall rates for a twenty-one year record of warm season precipitation. The fraction of hourly precipitation partitioned between infiltration and surface runoff is summarized storm event-wise across all parcels. Using a subset of the data ($n = 426$), we display uncertainty estimates on the fraction of precipitation that is infiltrated, for different hourly design storms. The 99% confidence intervals are based on a propagation of uncertainty through models used to estimate infiltration rates (Fig. S4, S5).

3. Buffalo is built largely on lake bed, with mixtures of sands, clays and some till. At least some of the clays are thixotropic. Relatively flat topography and continental climates can yield seasonally high groundwater levels which can reduce infiltration capacity. The model for infiltration capacity assumes free drainage as a lower boundary condition. This may result in overestimation of effective infiltration by underestimation of saturated areas.

Response: This is a great point. We investigated this and summarize existing observations of water tables across Buffalo in the SI of the revised manuscript. We show that Buffalo tends to have deep water tables interspersed with only a few locations with shallow water tables.

Overall, these comments (in addition to those from Reviewer 2) have prompted us to more clearly outline the assumptions of our model in the Supporting Information Section S8, and we demonstrate with figures that these assumptions largely hold across the city.

As stated in SI (Section 8):

“As a simple representation for how we conceptualize the movement of water through vacant land areas, the model makes several fundamental assumptions:

- 1) Vacant lots are gently sloped, thereby slowing the velocity of runoff at the surface and lateral water redistribution in the subsurface
- 2) Low infiltration rates were noted on lots with gentle slopes, such that localized overland flow is retained on the lot, and is not otherwise routed onto impervious surfaces, nor out of the lot,
- 3) Lots are well-vegetated, with vegetation providing resistance to the movement of water at the surface,
- 4) Groundwater table depth is well below the surface, diminishing the chance that runoff formation via saturation excess will occur, and
- 5) The near subsurface does not contain any restrictive layers that otherwise restrict redistribution of soil moisture and vertical percolation of infiltrated water. “

To further emphasize that these assumptions may not apply to all cities, we have included the following text in the Discussion section:

Page 17 lines 321 – 330: “Locally elevated water tables, which can occur seasonally (e.g., in proximity to local water bodies) and as a result of local soil conditions and drainage, may limit the efficacy of vacant lands as stormwater resources at sites with these characteristics. In the case of Buffalo, shallow water tables do exist in some locations, though water table depths across the city are typically deep (Fig. S8, Fig. S9). As our estimates do assume free drainage as a lower boundary condition, evaluating the sustainability of vacant lots to act as a stormwater resource would benefit by expanded assessments that account for both surface (such as the data here), and subsurface contributions to water cycling in cities⁴⁸. Subsurface architecture and associated microporous spaces are equally important controls on the detention capacity of vacant lots²⁴, but remain poorly understood in urban environments.”

Page 19-20 lines 367 – 381: “Vacant land may be a particularly important stormwater management resource in urban settings such as Buffalo. Here, lower overall rainfall depths and intensities favor infiltration over runoff. Yet, we advise that any concerted planning or demolition effort to treat vacant lands as a stormwater management tool should ultimately consider both macro- and micro-features of the landscape. Each city will have a unique combination of rainfall pattern, soil properties and associated heterogeneity, and vegetation that must be considered before any attribution of stormwater detention benefits is made to these spaces. Likewise, as these properties and context vary extensively from parcel to parcel, obtaining more detailed information is key, and one of the foundational drivers for making these assessments. This may be particularly true in terms of how soils are layered in the subsurface. A

comprehensive surface-subsurface assessment will help to identify parcels with the highest storage capacity, and ensure that those with the lowest capacity are relegated to other land uses, such as redevelopment. In the case where there are locally high water tables, the possibility of their regulation of subsurface storage potential is important, especially in terms of assuming runoff formation via infiltration- than saturation-excess mechanisms.”

4. Are parcels entirely residential or do they include former industrial lands? Brown fields and contaminated subsoils occur through different parts of Buffalo, including areas where vacant lands were sampled. How much of a potential groundwater contamination do the expanded vacant land pose?

Response: We leveraged a publicly available dataset of 16,000 vacant parcels to assess the former land use of currently vacant spaces. In Buffalo, nearly all vacant lots were formerly residential or commercial properties (see Table S1 below – page 3 of Supporting Information).

Table S1: Former land use indicated by property class description of Buffalo vacant lots (n = 16,119).

Land Use	All Properties n = 16,119			
	Total (#)	Area (ha)	Total (%)	Area (%)
Residential Vacant Land (311, 312)	13976	4.89	86.7%	65.0%
Commercial Vacant Land (330)	1641	1.38	10.2%	18.3%
Industrial Vacant Land (340, 341)	502	1.26	3.1%	16.7%

To emphasize these points, we added the following text to the Discussion Section:
 Added Text – Page 18 lines 344 – 351: “Stormwater benefits across vacant lots spell unclear impacts for disturbing or introducing contamination in the subsurface. A common perception associated with stormwater infiltration in vacant lots is that there may be increased risk from infiltrating contaminated water or mobilizing contaminant transport in the subsurface. Demolition can alter parcel microtopography; these changes to topography may introduce new pathways of surface water routing from roads to vacant lots, potentially carrying and infiltrating a host of contaminants carried within stormwater⁵⁰. Therefore, the balance between the hydrological benefits of vacant lots and potential water quality impacts should be considered.”

Reviewer #2 (Remarks to the Author):

This paper estimates the impact of vacant lots in Buffalo on infiltration rates. By doing so the paper quantifies an important potential benefit of these areas for flood risk and potentially groundwater recharge in this city.

Response: We thank the reviewer for their comments. Indeed, more than flood risk, we think these lots reduce water entering the stormwater system, with potential benefits for aging water infrastructure with finite capacity.

The paper could be a useful starting point for evaluating the ecosystem service contribution of vacant lots in humid environments. Using these results to support the parameterization of a hydrologic model of runoff generation (and other impacts of urbanization) would be a valuable extension of the work.

Response: We completely agree, and this is certainly an avenue we aim to explore in the future.

My main concern with this work is the assumption that infiltrated water necessarily contributes to detention capacity. If infiltrated water follows preferential flowpaths to surface or even soil moisture in down gradient areas, the effective detention may be quite minimal at hill slope to watershed scales.

Response: This is an interesting point, and one that we have thought a lot about in the context of this work. We will address this concern below using a broad lens, followed by a focus across the urban hydrology literature, and finally from local perspective, in the context of the City of Buffalo.

Broadly, the scientific community knows very little regarding preferential flowpaths in urban areas, and how water travels in the subsurface in these areas (Jefferson et al., 2017). We expect preferential flow paths in urban settings to be a function of topography (and local slope), the height of the water table, and soil properties. However, this is often impacted by buried infrastructure (and leaky pipes, especially), debris that can create preferential flowpaths and macropores (Shuster et al. 2014; Landscape and Urban Planning), and even the presence of basements and other built structures (Bhaskar et al., 2016). (Note, however, that local demolition regulations require contractors to remove all debris from the site and backfill with clean soil fill, which may therefore remove one source of structural macroporosity.)

Given these uncertainties, the reviewer is correct that infiltrated water could follow preferential flowpaths to the surface in some areas, but we expect that these effects are likely to be highly localized, occurring over only a small fraction of properties. This is particularly true for Buffalo because it is very flat. Buffalo varies by – at most – 135 meters from its highest point to its lowest point. These exceptionally low land surface slopes do not favor rapid subsurface water movement. Furthermore, the relatively low slopes of each vacant lot grade into an impervious surface at the front of the lot, e.g., a sidewalk, that extends into the soil profile. This would block macropore flow from the lot further downgradient (at least to the depth that the impervious surface extends into the soil profile).

While there is a possibility that dynamic water tables could transmit subsurface water from vacant lots to other areas, we expect that nearly all of this water will leave these areas as evapotranspiration because of the low slopes and vegetation on the vacant lots. With most storms lasting < 24 hours at low rainfall intensities, it is unlikely that preferential flow is a driver of subsurface flows in this system.

The reviewer's comment emphasizes the importance of considering the role of local conditions when designing any form of stormwater adaptation. It also underscores how little we know regarding the subsurface movement of water in urban areas (and in most systems), and that this

is a sorely needed area of future research. In response to the reviewer's concern, we have emphasized these two points in the updated manuscript:

Page 17 lines 325 - 330:

“As our estimates do assume free drainage as a lower boundary condition, evaluating the sustainability of vacant lots to act as a stormwater resource would benefit by expanded assessments that account for both surface (such as the data here), and subsurface contributions to water cycling in cities⁴⁸. Subsurface architecture and associated microporous spaces are equally important controls on the detention capacity of vacant lots²⁴, but remain poorly understood in urban environments.”

Page 19-20 lines 367 – 381: “Vacant land may be a particularly important stormwater management resource in urban settings such as Buffalo. Here, lower overall rainfall depths and intensities favor infiltration over runoff. Yet, we advise that any concerted planning or demolition effort to treat vacant lands as a stormwater management tool should ultimately consider both macro- and micro-features of the landscape. Each city will have a unique combination of rainfall pattern, soil properties and associated heterogeneity, and vegetation that must be considered before any attribution of stormwater detention benefits is made to these spaces. Likewise, as these properties and context vary extensively from parcel to parcel, obtaining more detailed information is key, and one of the foundational drivers for making these assessments. This may be particularly true in terms of how soils are layered in the subsurface. A comprehensive surface-subsurface assessment will help to identify parcels with the highest storage capacity, and ensure that those with the lowest capacity are relegated to other land uses, such as redevelopment. In the case where there are locally high water tables, the possibility of their regulation of subsurface storage potential is important, especially in terms of assuming runoff formation via infiltration- than saturation-excess mechanisms.”

Saturation from below may also be an issue and this was not addressed in the paper. In other words, this study presents convincing evidence that infiltration rates are high in vacant lots - but does not necessarily show that these high infiltration rates provides substantial hydrologic services. Further analysis is needed before concluding that improved infiltration leads to meaningful increases in “detention capacity” that reduce runoff at a city scale.

Response: Saturation from below is certainly a possible issue that would limit the potential of vacant lots to infiltrate rainfall. To address this, we conducted an in-depth literature investigation and analyses of water tables across Buffalo. Specifically, we first amalgamated water table observations from the USGS over the past 50 years (Fig. S8). While many of these observations are from single points in time, we next conducted an analysis of recent data (Fig. S9), which suggest that water tables in Buffalo are typically deep - though some locally shallow (high) water tables do exist. We therefore posit that outside of potentially (and likely sparsely) interspersed locally high water tables, our assumption that infiltration rates translate to increases in detention capacity is valid. We emphasize these findings in the revised manuscript, along with the importance of considering water table depths in any plans or assessments of potential stormwater benefits of vacant lots.

We added text indicating this in paragraph 4 of the Discussion in the revised manuscript, noting that (Page 17 lines 321 – 330) “Locally elevated water tables, which can occur seasonally (e.g., in proximity to local water bodies) and as a result of local soil conditions and drainage, may limit the efficacy of vacant lands as stormwater resources at sites with these characteristics. In the case of Buffalo, shallow water tables do exist in some locations, though water table depths across the city are typically deep (Fig. S8, Fig. S9). As our estimates do assume free drainage as a lower boundary condition, evaluating the sustainability of vacant lots to act as a stormwater resource would benefit by expanded assessments that account for both surface (such as the data here), and subsurface contributions to water cycling in cities⁴⁸. Subsurface architecture and associated microporous spaces are equally important controls on the detention capacity of vacant lots²⁴, but remain poorly understood in urban environments.”

We also added our analysis to the Supporting Information, as described below.

“SI Section S8 (pages 14 – 16 in Supporting Information):

To assess whether saturation overland flow may limit the potential of vacant lots to infiltrate precipitation, we gathered spatially distributed and temporally resolute groundwater observations from across Buffalo. Though direct observations of water table depths are often limited, we queried the United States Geological Survey database to locate one-time observations of water table depth across the city. If multiple observations were taken at a location, we selected the observation closest to the surface. These water table depths are visualized in Figure S8 below. While many water table depths are many meters below the surface, four observations across the city, representing 17% of sites, suggest that the local water table is within 0 to 0.3 m of the surface. We do note that these observations were taken nearly 50 years ago, but still represent a source of information regarding the presence of locally perched water tables.

Figure S8: Instantaneous groundwater levels as measured by the United States Geological Survey across *Buffalo*.

In contrast, timeseries observations from a newly installed well (est. 2017) on the State University of New York Buffalo campus (Fig. S9), suggest that the (local) water table is deep and varies minimally throughout the year. Taken together, water table timeseries and spatially distributed observations do suggest that high water tables may occur across the city of Buffalo, but that these occurrences are few in comparison to the number of deeper water tables. While derived from only one site, observations do hint that water table depths through the year may vary less than a meter between wetter times (winter and spring) and drier times (summer and fall).

Fig. S9: Timeseries of water table depth from USGS groundwater well (#430006078464101; Local number: E-1929) located on SUNY Buffalo campus.”

Another limitation is the lack of comparison with alternative land covers. Comparison with other types of lots - low density housing etc. is also needed before “additional infiltration” can be quantified. It is not clear in the study what the ‘null’ state is - what would be the infiltration of a “developed” lot in the neighborhoods with the vacant lots.

Response: This limitation was identified by multiple reviewers and has been the main focus of our re-analysis efforts (please also see the first two detailed responses to Reviewer 1). To estimate a ‘null state’ for vacant properties and to fully quantify the potential benefits of infiltration across these areas, building footprints on 2482 properties were hand delineated based on orthoimagery from 2002. We also estimated a ‘future state’ (“redevelopment”) using orthoimagery from 2017. We then simulated infiltration volumes for pre-demolition (“null”), post-demolition, and redevelopment (“future”) states using our parsimonious infiltration model (detailed in the Methods section of the manuscript). Together, these analyses were used to reconstruct Figure 4, which quantifies how infiltrated volume versus runoff volume change across these states. As we now demonstrate, despite potential redevelopment, infiltrated volumes show a net gain of between 42 and 66% on vacant lots when compared to infiltration volumes on the 2482 pre-demolition/null state parcels.

It may also be important to consider that there may be water quality issues associated with water moving through potentially contaminated soils in these vacant areas. These effects may indeed be small - these types of additional issues should at least be acknowledged in the discussion.

Response: In response to this comment, we expanded our recommendations to include an improved discussion of potential impacts of vacant parcel demolition and redevelopment on subsurface contamination.

This text is included on Page 19 lines 344 – 351: “Stormwater benefits across vacant lots spell unclear impacts for disturbing or introducing contamination in the subsurface. A common perception associated with stormwater infiltration in vacant lots is that there may be increased risk from infiltrating contaminated water or mobilizing contaminant transport in the subsurface. Demolition can alter parcel microtopography; these changes to topography may introduce new pathways of surface water routing from roads to vacant lots, potentially carrying and infiltrating a host of contaminants carried within stormwater⁵⁰. Therefore, the balance between the hydrological benefits of vacant lots and potential water quality impacts should be considered.”

There are some interesting results - related to characterizing vacant parcels - In particular the finding that most are “well-vegetated, gently sloped” - and the finding that exceptions tend to be “near roadways where demolition activities and traffic area concentrated” - A key question, however, is how generalizable are these findings to other cities. A more thoughtful assessment of this generalizability is needed. For example, on line 308 authors note that vacant land may be particularly important in areas where rainfall depths/intensities are lower than in Buffalo – but these regions may also have substantial differences in soil and vegetation which may change the infiltration rates of vacant lots!

Response: Based on these recommendations, we have expanded Discussion text in the revised manuscript to include the following:

Page 19-20 lines 367 – 381: “Vacant land may be a particularly important stormwater management resource in urban settings such as Buffalo. Here, lower overall rainfall depths and intensities favor infiltration over runoff. Yet, we advise that any concerted planning or demolition effort to treat vacant lands as a stormwater management tool should ultimately consider both macro- and micro-features of the landscape. Each city will have a unique combination of rainfall pattern, soil properties and associated heterogeneity, and vegetation that must be considered before any attribution of stormwater detention benefits is made to these spaces. Likewise, as these properties and context vary extensively from parcel to parcel, obtaining more detailed information is key, and one of the foundational drivers for making these assessments. This may be particularly true in terms of how soils are layered in the subsurface. A comprehensive surface-subsurface assessment will help to identify parcels with the highest storage capacity, and ensure that those with the lowest capacity are relegated to other land uses, such as redevelopment. In the case where there are locally high water tables, the possibility of their regulation of subsurface storage potential is important, especially in terms of assuming runoff formation via infiltration- than saturation-excess mechanisms.”

We agree that it is challenging to generalize our results; instead, we aim to generalize how the information we have obtained could be used in other cities to repurpose vacant lands and to holistically evaluate sources of stormwater abatement that extend beyond green infrastructure alone. This point is emphasized as well on page 20-21 lines 393-404:

“Many cities are alive with green spaces that are valued for numerous ecosystem services^{49, 51-53}. In this way, neighborhoods may consider a portfolio approach to stormwater management, interspersing residential lawns, green infrastructure, and vacant landscapes to create greater connectivity of green spaces, likely with numerous benefits for city inhabitants when distributed

equitably. While not considered in our analysis, future work may also consider the role of residential lawns alongside vacant lands. We currently have little to no baseline for comparison regarding how infiltration rates on vacant lots compare with nearby residential lawns. Yet, if these residential green spaces are maintained properly, these areas are likely to provide additional capacity for stormwater volume, beyond those attributed to vacant lands. Many cities already take a ‘portfolio’ approach to planning and installing different sizes and types of GI. Adding vacant lands and even residential lawns to these planning efforts may be a natural extension of these efforts and offer greater flexibility¹⁵.

Finally soil characteristics are notoriously variable at fine spatial scales. Some assessment of variation in infiltration (as measured with the infiltrometer) using multiple samples for a few sites would provide error bounds on the mean parcel estimates. Estimates of uncertainty would strengthen this work. Are two sample per lot sufficient?

Response: We do agree that assessments of uncertainty would be useful for this analysis. While completing additional infiltration tests would be one way to assess uncertainty, this dataset was collected more than three years ago; therefore, comparing any new estimates of infiltration would not be comparing ‘like’ with ‘like’. We strongly believe that infiltration is an emergent and time-varying property of vacant lands. Re-estimating infiltration rates on any of these properties would allow us to estimate infiltration at the current state, but this is not necessarily comparable to infiltration rates from three years past. Differences in infiltration rates would instead be attributable to any host of influences acting on vacant parcels over the past three years. As these lands continue to rebound, we expect that their infiltration rates will evolve and change, albeit in different ways depending on degree of saturation, weather, soil type, human activity, and any redevelopment.

In response to this comment, rather than investigating uncertainty through repeat assessments, we have propagated an estimate of measurement error through our infiltration model. This measurement error is a likely source of uncertainty within our dataset and has great potential to reveal the impact of uncertainty on our analysis. To assess how uncertainty translates from a single water level measurement to an estimate of infiltration, we have quantitatively propagated an uncertainty in water level observation of 1 mL, which was added to and subtracted from each time point, one at a time, for each infiltration test. These assessments of uncertainty are shown in a revised Figure 1, and in Supporting Information (Fig. S4, S5). These uncertainties were also propagated through estimates of infiltrated volume across hourly design storms for the subset of properties (n = 426) with documented timeseries of infiltrated volume. This is described in Supporting Information, Section S7: Uncertainty in infiltration estimates. We include two figures from this section below (Fig. S4, Fig. S5).

Fig. S4: Mean (circles) and 99% confidence intervals (lines) of infiltration rates from propagating measurement uncertainty (± 1 mL) through timeseries of infiltrometer water levels.

Fig. S5: Mean infiltration rate and 99% confidence intervals (y-axis) as compared to measured infiltration rate ($n = 426$).

We then addressed the question of whether our measurements are representative of each lot. To do this, we resampled the lot-averaged infiltration dataset to assess whether the mean of this dataset is robust from scaling to the larger vacant lot assessment. Our analysis confirmed that the mean is consistent (Figure S6).

line 103-104- "different ranges of infiltration rates were co-located" - this is too vague to be meaningful

Response: We updated this to: “Notably, parcel infiltration rates within all sewersheds varied by between three and four orders of magnitude (Fig. 1).” (page 7 lines 120 – 121)

line 107 - the sentence structure implies that the lack of a relationship between demolition type and infiltration is somehow tied to the increase in demolition. - I don't think this is what is intended

Response: We updated this sentence to clarify this is not the case.

line 157 Some statistics on these relationships (that support the conceptual model) would be lend credibility here (e.g what is the relationship between vegetation cover and estimated infiltration).

Response: We clarified our meaning in these statements by updating this text in the revised manuscript. Our meaning was not to suggest that vegetation cover is responsible for estimated infiltration rates (and, conversely, that we should expect low infiltration rates in poorly vegetated lots). Instead, our goal was to demonstrate that infiltration rate is one of many characteristics across vacant lots that may may suppress runoff. We updated this text to the following (page 10 lines 178 – 185):

“While infiltration rate determines how quickly any surface water will move into the soil, the form of vacant lots may also shape the function of these areas to suppress runoff and favor infiltration. Beyond infiltration rates alone, we found that several vacant lot characteristics likely promote rainfall detention capacities and suppress runoff, namely: the presence of vegetated land cover, minimal soil compaction in the greater extent of the parcel, and low parcel slopes. The characteristics that support our conceptual model also indicate that vacant lots display a common internal soil and landform structure that can act to suppress runoff production, through heightened rainfall detention capacity.”

Given the data some additional information on types of vegetation cover might be useful to analyze (or at least describe what types of vegetation are typically found in vacant lots)

Response: Initially, lots were planted with a mixture of fescue and white clover. All lots subject to the full assessment (n = 718) were also surveyed for the presence of common invasives, including phragmites, knotweed, garlic mustard, and loosestrife. While knotweed and garlic mustard were prevalent at the time, phragmites and loosestrife were less common (see Table X). Other invasives observed on lots across the city include common lambsquarter and crabgrass.

Table 2. Prevalence of common invasives on vacant lots (n = 718).

	Phragmites	Knotweed	Garlic Mustard	Loosestrife
Lots (#)	13	187	184	12
Lots (%)	2%	26%	26%	2%

This information is now noted on page 10 lines 188 – 194:

“Vacant parcels, following demolition, were initially planted with a mixture of fescue and white clover, though common species observed across vacant lots included crabgrass (*Digitaria spp.*), knotweed (*Fallopia japonica*; present on 26% of fully assessed parcels), garlic mustard (*Alliaria petiolata*; present on 26% of fully assessed parcels), and common lambsquarter (*Chenopodium album*). These ruderal plants aggressively establish and spread, and effectively contribute to increasing the vegetative cover in vacant parcels. Their success will also impart surface roughness, which is expected to limit further routing of localized ponding or runoff production.”

Reviewer #3 (Remarks to the Author):

The authors combine an impressive field dataset and simple hydrological models to demonstrate the potential of vacant land to contribute to green infrastructure. The work represents among the most thorough assessments of vacant lots I have seen to date, and the manuscript is well researched and clearly presented. It will make a useful contribution to the urban ecology and hydrology literature. I have only very minor suggestions.

Response: We appreciate your feedback and encouragement.

Abstract:

The abstract would benefit from more clarity re: the relationship of the present findings to demolition techniques and regulations. Specifically:

Line 28: “Despite variation in demolition technique”. This line is unclear. Wouldn’t one expect infiltration to span different rates with variation in demolition technique? Clarify whether infiltration is/isn’t correlated with variation in demolition technique here.

Response: This has been clarified in the abstract and results sections.

In the abstract, we now state: “Vacant lot infiltration spanned low (0.001 cm hr⁻¹) to high (5.39 cm hr⁻¹) rates, and was not correlated with variation in demolition age or technique.”

In the results, also clarify: “Parcel-level infiltration rates ranged from 0.001 to 5.39 cm hr⁻¹ (n = 520; mode: 0.001 cm hr⁻¹) across fourteen sewersheds (Fig. 1; Fig. S1). When compared across the city, parcel-level infiltration rates had weak spatial structure (Fig. 1; Table S5 – S6).”

Line 32: Unclear what the role of regulation/demolition process is, here. In what way should it be regulated? What effect would this have?

Response: Given the limited word count of the abstract, we removed the reference to the demolition process in the revised manuscript, and saved this for the discussion section instead.

Introduction:

Overall, the introduction is clear and well written. Given the structure of the journal, the

introduction would be improved by a minor increase in the methodological details provided (e.g. lines 90-93) so that the reader can better interpret the results in the following section. While this information is clearly presented in the supplemental information, one or two sentences within the introduction clarifying, e.g., which soil properties were measured and whether/how field methods were used would improve the flow of the manuscript.

Response: Based on these recommendations, we revised the introduction to (page 5-6 lines 96 – 109):

“Each of 520 unique parcels were assessed using a standard protocol that yielded qualitative and quantitative data characterizing land cover (nine observations per site), soil type (two observations per site), topography (nine observations per site), compaction (four observations per site), former building footprint (from historical orthoimagery), and infiltration rates (two observations per site). To document the potential benefit of urban vacant parcels for stormwater management, we applied a simple infiltration-excess model incorporating estimates of perviousness with four key goals, which include: (1) to test for spatial structure in rainfall detention capacity, (2) to determine sets of threshold rainfall events that initiate runoff from these properties, (3) to identify classes of vacant lots that most effectively infiltrate rainfall, and (4) to quantify changes in detention capacity provided by vacant land at city-wide scales. Our inferences are drawn by extending a smaller dataset (n = 520) to ca. 2400 vacant parcels across Buffalo, and contextualized with pre-demolition and recent building footprints to simultaneously document the evolution of vacant lands while quantifying the impact of this evolution on hydrologic partitioning at the land surface.”

References:

Bhaskar, A.S., Beesley, L., Burns, M.J., Fletcher, T.D., Hamel, P., Oldham, C.E. and Roy, A.H. Will it rise or will it fall? Managing the complex effects of urbanization on base flow. *Freshwater Science*. 35(1), 293-310 (2016).

Jefferson, A. J., et al. Stormwater management network effectiveness and implications for urban watershed function: A critical review. *Hydrol. Process*. **31**, 4056 – 4080 (2017).

Masoner, J. R., Kolpin, D. W., Cozzarelli, I. M., Barber, L. B., Burden, D. S., Foreman, W. T., ... & Hopton, M. E. Urban Stormwater: An Overlooked Pathway of Extensive Mixed Contaminants to Surface and Groundwaters in the United States. *Environmental Science & Technology*. 53(17), 10070-10081 (2019).

Reviewers' comments:

Reviewer #1 (Remarks to the Author):

This is an intriguing paper as it deals with an intersection of major urban challenges - stormwater mitigation and vacant land. While set in Buffalo, NY, the implications are important for a number of cities with similar circumstances, now facing decisions on how they can rebuild, and what the trade-offs are.

The authors have responded to my initial comments and have included additional analysis, text and graphics to address these concerns. There are a few minor areas that should be addressed:

1. I asked if the volumes of stormwater reductions (given in cubic meters) could be expressed as an equivalent depth so that it can be understood within the context of the total precipitation (given in figure 4 in cm). This would answer the question of how much of the total input of precipitation is now being diverted to infiltration vs. runoff in the same units. This is a minor addition, just requiring division by the appropriate area (perhaps the full City of Buffalo).

2. The question of the free drainage assumption for the infiltration estimates (deep groundwater table) was addressed by pulling (largely) one time groundwater depth from wells taken in a USGS 1960s survey. At this time, Buffalo was at its peak build out - serious decline, arson and housing abandonment began in the 1970s. It may be that this time corresponded with the greatest impervious runoff, and consequent deeper groundwater tables. The Bhaskar paper cited by the authors acknowledges that deeper or shallower groundwater tables are a function of both hydrogeology and the urban structure, with areas with extensive open ground potentially having higher groundwater tables. If groundwater recharge has increased significantly in the areas of vacant land, can this produce local groundwater mounding, compromising the free drainage assumption? The 1960s groundwater measurements may not answer this question so it would be prudent to include this discussion. The one current groundwater well on the SUNY@Buffalo campus needs more context to be interpreted - Where is it? On the old campus in the city, or out in Amherst? If on the old campus (Main Street and Bailey Ave), that area is build on a paleo-lakeshore with sandier soils and presumably deeper groundwater levels than most of the city which is build on flatter paleo lakebed. This should simply have more discussion to help the reader understand the context.

3. SSURGO information often does not incorporate the effects of urbanization on soils - often soils are referred to as "urban complex." While work elsewhere supports the findings that urban soils can have high infiltration rates, acknowledging the limitations of SSURGO in urban areas is prudent.

I think these are easily addressed points which should improve the paper. The paper's analysis and findings provide an important set of concepts and proposals to publish, and could be influential on urban planning, particularly as distressed cities contemplate how to recover and rebuild from urban decline. A priority is to rebuild the tax base which was severely eroded by population and residential decline - the tradeoff of returning to densified development needs to be understood.

Larry Band

Reviewer #2 (Remarks to the Author):

The current version of this paper has addressed some of the concerns that I identified in my initial review. However much of the presentation of these ideas is challenging to follow - and make the paper difficult to read. While I understand that the details of methods are provided in the Methods section at the end, the general approaches used should be readily understandable from reading

the main body of the paper - I point out a number of example below in my detailed comments. Here is an example, the question of what happens to infiltrated water was addressed on line 325-330 and 367-381 - but the writing still obscures the point - (or assumes substantial hydrologic knowledge on the part of the reader to interpret)

eg. "As our estimates do assume free drainage as a lower boundary condition, evaluating the sustainability of vacant lots to act as a stormwater resource would benefit by expanded assessments that account for both surface (such as the data here), and subsurface contributions to water cycling in cities⁴⁸."

This sentence is long - and does not explain what the implications of the free drainage is with respect to conclusions about storm water detention made in this study. - a more direct statement of the limitations associated with the assumption of free drainage is needed here.

Perhaps some clarity could be provided by introducing the basic conceptual model up front - that clearly shows assumptions in how you are assessing vacant land impacts on storm water retention (perhaps a figure to illustrate your terminology)

The discussion section is also now overly lengthy - Some careful re-editing to shorten and clarify the key concepts would be helpful . The paper is also within the domain of green infrastructure and urban green spaces, which has become a mature literature - thus placing this paper into that literature, particularly in the discussion, would help to make conclusions more general. Based on current literature, how distinct are vacant lots from urban green space (lawns) etc., how do estimates here compare with other model and measured estimates of urban green space hydrologic benefits in other locations?

Abstract 28 - 'present' is awkward - present what?

line 34 - not sure how 'comparing pre- and post demolition partitioning' fits into this sentence, partitioning of what? What are you assuming about pre-demolition here? Re-word for clarity.

line 35 - I don't see how these finding differentiate vacant lots as 'multi-functional' landscapes. First the authors have not really made the case that vacant lots are multi-functional - only that that infiltrate water AND it is not clear that other types of land use are not multi-functional as well.

line 47 - In some places the redevelopment value of vacant land maybe significant....include citation?

line 59 - I don't see how rainfall patterns (at the time scale relevant to detention capacity) force surface landform and soil properties

line 75 - how is accounting for distributed presence (of what?) a strategy? This wording does not communicate what I think the authors are trying to say

line 79 - you can't really posit that cities have large portfolios (you aren't testing this posit)- and some cities don't have large areas of vacant lots- Rather in cities that do have large portfolios of decentralized pervious surfaces there are opportunities.

line 81 - I'm not sure what is meant by footprint here - this implies that the amount of urban

vacant land is growing and I don't see evidence of this in your background

line 75-87 - this is a key paragraph for setting up the background - but needs some more thinking to make sure that the authors do not over state the case - and the meaning of sentences is clear...

line 95 - at this point in the paper, the narrative switches to a single city - Buffalo. An analysis using a single study site is fine - but it needs some introduction - why is Buffalo a good place to do this analysis? What characteristics of this city make it a good example of the kind of potential vacant land benefits that have been discussed thus far.

line 101 This model is central to your analysis - a bit more upfront description here would help readers understand your results - just 1-2 sentences.

line 103 - what is meant by a 'set' here - rainfall events in sequence?

line 107-108 - how are vacant lands evolving? (implies the vacant lands are somehow changing?) - it is also unclear what contextualized means here - and what is meant by a recent building footprint - recent in what way?

line 122 - - 'that' has an unclear antecedent - rephrase

line 127 - Some discussion of why infiltration rates can be a reasonable proxy for detention capacity is needed here - a capacity is typically a maximum potential storage value
So there are two points here that need justification 1) is it reasonable to assume that all infiltration is retained as storage (rather than lost through drainage) and 2) presumably infiltration rate varies with rainfall intensity - a capacity suggests a maximum value - so clarify if maximum infiltration rates used here. If so how were they determined.

Fraction of precipitation that is infiltrated is much more clearly defined (e.g as shown in Figure 2 - using the term 'rainfall detention capacity' adds confusion -

line 144-147 this explanation is hard to follow - what does it mean to employ infiltration rate across pervious fractions -

How is parcel 'condition' summarized (what is condition)

line 146 - what is the 'infiltration-excess processes' referred to here - description of the approach needs to be more precise - Equations would be helpful

line 156-157- how is uncertainty computed? (e.g uncertainty due to what?)

line 161 aggregated over what?

line 182 - what is meant by the 'the greater extent of the parcel' greater than what?

line 177 include the statistical method that was used to identify relationships between infiltration rates and characteristics noted.

line 196 - was it slow that averaged below 1.5%?

line 200-203 while I agree that all of these features likely matter (and most simple conceptual models of hydrologic response) would agree, this discussion is overly vague. How does the gentle slope increase water detention (I would expect the opposite)

A key question is how runoff is partitioned between surface detention store and transpiration fluxes that lead to water returning to the atmosphere versus drainage - either rapid or shallow subsurface - It would be helpful to provide some insight into this

Figure 3 - how specifically do these features shape infiltration (e.g what analysis was done to support this statement)

line 224 - is there data suggesting that vacant land in Buffalo is expanding?

line 228 'representative assessment of average pace hydrology' is not clear -explain what this is - do you mean average maximum infiltration rate?

line 230 - provide some indication of how this estimate of footprints was made - using what data and assumptions (and time period over which removal occurred)

line 233 how were rates of redevelop and demolition determined?

line 246 - clarify what the baseline is here - increased relative to what?

line 267 - it would helpful to be more precise here - what data is used to estimate the thousands of parcels?

line 264 - I don't really understand the point here - if building footprints are small - then vacancy many not significantly add to expected infiltration - e.g if imperviousness is small

Could the authors provide a simple estimate of the range in pervious cover associated with developed parcels - and then use that quantitative estimate in the comparisons?

I think the authors have done this- but it is challenging to extract this information from the paper as written

line 277 - this was true ONLY in Buffalo - The sentence implies a generality that is not there
So in Buffalo, we found that...

line 293-295 there are other studies that document the role of vegetated patches in urban environments on reducing urban runoff - it would be useful to place estimates found here into this broader literature context.

line 314 - what may be underestimated?

line 317 - meaning of landscape or system "storage versus their respective capacities" is unclear

line 335 this is too vague and mixes concepts of sustainability and equity - without meaningfully describing the relationship with either concept

Dear Reviewers,

We thank you for your feedback and have responded to each of your comments inline below. Thank you for your assistance in strengthening our manuscript.

Reviewer #1 (Remarks to the Author):

This is an intriguing paper as it deals with an intersection of major urban challenges - stormwater mitigation and vacant land. While set in Buffalo, NY, the implications are important for a number of cities with similar circumstances, now facing decisions on how they can rebuild, and what the trade-offs are.

The authors have responded to my initial comments and have included additional analysis, text and graphics to address these concerns. There are a few minor areas that should be addressed:

1. I asked if the volumes of stormwater reductions (given in cubic meters) could be expressed as an equivalent depth so that it can be understood within the context of the total precipitation (given in figure 4 in cm). This would answer the question of how much of the total input of precipitation is now being diverted to infiltration vs. runoff in the same units. This is a minor addition, just requiring division by the appropriate area (perhaps the full City of Buffalo).

Based on this feedback, we have converted units in Figure 4 to units of depth instead of volume. We have moved the figure displaying volume estimates to our Supporting Information Section (Figure S12).

2. The question of the free drainage assumption for the infiltration estimates (deep groundwater table) was addressed by pulling (largely) one time groundwater depth from wells taken in a USGS 1960s survey. At this time, Buffalo was at its peak build out - serious decline, arson and housing abandonment began in the 1970s. It may be that this time corresponded with the greatest impervious runoff, and consequent deeper groundwater tables. The Bhaskar paper cited by the authors acknowledges that deeper or shallower groundwater tables are a function of both hydrogeology and the urban structure, with areas with extensive open ground potentially having higher groundwater tables. If groundwater recharge has increased significantly in the areas of vacant land, can this produce local groundwater mounding, compromising the free drainage assumption? The 1960s groundwater measurements may not answer this question so it would be prudent to include this discussion.

The one current groundwater well on the SUNY@Buffalo campus needs more context to be interpreted - Where is it? On the old campus in the city, or out in Amherst? If on the old campus (Main Street and Bailey Ave), that area is built on a paleo-lakeshore with sandier soils and presumably deeper groundwater levels than most of the city which is built on flatter paleo lakebed. This should simply have more discussion to help the reader

understand the context.

This is a great point. The well on campus is indeed located in till moraine while the majority of the lots are located in lacustrine silt and clay surficial geology.

We have clarified this on page 19 (lines 356 – 369): “This assumption will be violated in areas where soil moisture is concentrated or forms a water table (e.g., localized groundwater mounding), predisposing the landscape to generate runoff by saturation-excess⁵⁵, or form variable source areas⁵⁶. Though the majority of Buffalo is located in lacustrine silt and clay surficial geology, shallow water tables were historically observed in only a few locations, and water table depths across the majority of the city were typically deep (Fig. S8). As little is known about the urban subsurface in Buffalo and other urban areas, we recommend expanded, standardized field assessments to catalog and categorize vacant parcels, including their subsurface hydrology¹⁷. Such assessments will provide training datasets that improve the representativeness and generalizability of simulation models that scale vacant lot detention capacities to cities. The assessments would also refine our understanding of locations in the urban landscape that contribute to surface runoff generation²⁴ – and how these source areas spatially vary and temporally interact with changes in local precipitation patterns to alter the mechanisms, frequency, and magnitude of urban runoff generation.”

We have also clarified this in the Supporting Information:

Section S8, page 14:

“While many water table depths are several meters below the surface, four observations across the city, representing 17% of sites, suggest that the local water table is perched to within 0 to 0.3 m of the surface. Notably, the majority of these water table observations are located in a low permeability lacustrine silt and clay surficial geology. These findings suggest that locally high water tables can occur across Buffalo (Fig. S8), but that many local water tables will still be relatively deep.”

Section S8, page 14-15:

“Though no timeseries observations of water table depths exist within the city, timeseries observations were available at a newly installed well (est. 2017) on the State University of New York Buffalo campus (Fig. S9). These observations suggest that the (local) water table remains at a relatively constant 6 meters depth below ground surface. Notably, this well is located in a till moraine, and expected to have greater permeability and therefore lower water tables than those in lacustrine silt and clay. Therefore, we expect vacant lots on the edges of the city to have low and consistent water tables that should not restrict any assumptions of free drainage.”

3. SSURGO information often does not incorporate the effects of urbanization on soils - often soils are referred to as "urban complex." While work elsewhere supports the findings that urban soils can have high infiltration rates, acknowledging the limitations of SSURGO in urban areas is prudent.

We completely agree. In fact, SSURGO’s limitations in urban settings was a motivation for performing this analysis. Co-authors on this article are doing much of the groundbreaking work

to demonstrate these limitations in urban areas. We follow the method as published in Herrmann, et al. (2018):

Herrmann, D.L., L.A. Schifman, W.D. Shuster. 2018. Widespread loss of intermediate soil horizons in urban landscapes. Proc. Natl. Acad. Sci. <https://doi.org/10.1073/pnas.1800305115>.

To further clarify how we have incorporated SSURGO soils, we have made the following adjustments to the SI material (Section 13):

Addition 1 – the first sentence of Section 13 has been updated to: To determine how reference soils, defined as soil complexes that have not been impacted by humans, compared with urban soils, we extracted information for pre-urbanization, reference soils from the Soil Survey Geographic Database (SSURGO), which is the best-known source for this type of data.

Addition 2: We undertook this analysis to compare vacant lot soil characteristics to the best-known (USDA) reference information, so that we can then contextualize how urbanization has altered local Buffalo NY soils. However, unless the county soil survey has been recently updated (e.g., Wayne County MI, USA) to account for new and modified soil series brought about by urbanization, the use of SSURGO soils data to represent urban areas is not recommended.

To further emphasize this point, we have added the following to page 13-14 lines 246 – 255: “In urbanized areas, characterizing reference, pre-urban land use and soils can be difficult. Similar to earlier work⁴³ and to address this challenge, we compared common urban soil units and their reference soils, available from the Soil Survey Geographic Database (SSURGO). This comparison benchmarks how our assessment compares with SSURGO estimates, which serve as the best information available for Buffalo, and other cities⁴³. As we show, inflated infiltration rates from these reference soil units would lead to large overestimates of detention volume (Fig. S15; Table S14). This circumstance underscores the value of high resolution field observations to inform the critical planning stages of re-development, and ongoing efforts to manage runoff. Our approach provides a robust starting point for science-based planning and infrastructure-asset needs assessments.”

I think these are easily addressed points which should improve the paper. The paper's analysis and findings provide an important set of concepts and proposals to publish, and could be influential on urban planning, particularly as distressed cities contemplate how to recover and rebuild from urban decline. A priority is to rebuild the tax base which was severely eroded by population and residential decline - the tradeoff of returning to densified development needs to be understood.

Larry Band

Many thanks for your thoughtful feedback, Dr. Band.

Reviewer #2 (Remarks to the Author):

The current version of this paper has addressed some of the concerns that I identified in my initial review. However much of the presentation of these ideas is challenging to follow - and make the paper difficult to read. While I understand that the details of methods are provided in the Methods section at the end, the general approaches used should be readily understandable from reading the main body of the paper - I point out a number of example below in my detailed comments.

We appreciate your feedback. We have made substantial changes throughout the manuscript to clarify the intent of our presentation and to convey the context and value of our work early in the manuscript, as you suggested. While we aim to keep with the style of the journal, for example, retaining the majority of the methods at the end of the manuscript, we have added more context on our methods to the final, scoping paragraph of our introduction.

Here is an example, the question of what happens to infiltrated water was addressed on line 325-330 and 367-381 - but the writing still obscures the point - (or assumes substantial hydrologic knowledge on the part of the reader to interpret)

eg. "As our estimates do assume free drainage as a lower boundary condition, evaluating the sustainability of vacant lots to act as a stormwater resource would benefit by expanded assessments that account for both surface (such as the data here), and subsurface contributions to water cycling in cities⁴⁸." This sentence is long - and does not explain what the implications of the free drainage is with respect to conclusions about storm water detention made in this study. - a more direct statement of the limitations associated with the assumption of free drainage is needed here.

We understand the reviewer's point and have revised this and the following sentences to (pg 19 lines 353 – 364):

“For example, we did not directly incorporate vacant lot subsurface architecture and associated macroporous spaces into our analysis, although these characteristics can influence the dynamic distribution of soil moisture in the soil profile²⁴. We also assume free drainage as a lower boundary condition. This assumption will be violated in areas where soil moisture is concentrated or forms a water table (e.g., localized groundwater mounding), predisposing the landscape to generate runoff by saturation-excess⁵⁵, or form variable source areas⁵⁶. Though the majority of Buffalo is located in lacustrine silt and clay surficial geology, shallow water tables were historically observed in only a few locations, and water table depths across the majority of the city were typically deep (Fig. S8). As little is known about the urban subsurface in Buffalo and other urban areas, we recommend expanded, standardized field assessments to catalog and categorize vacant parcels, including their subsurface hydrology¹⁷.”

Perhaps some clarity could be provided by introducing the basic conceptual model up front - that clearly shows assumptions in how you are assessing vacant land impacts on storm water retention (perhaps a figure to illustrate your terminology)

We have added more information clarifying our model framework on page 6 (lines 109 – 112):
“To document the potential benefit of urban vacant parcels for stormwater management, we

applied a simple infiltration-excess model. This model partitioned precipitation inputs between infiltration and runoff based on averaged, measured parcel infiltration rates (estimated as maximum infiltration rates) and pervious lot area.”

The discussion section is also now overly lengthy - Some careful re-editing to shorten and clarify the key concepts would be helpful . The paper is also within the domain of green infrastructure and urban green spaces, which has become a mature literature - thus placing this paper into that literature, particularly in the discussion, would help to make conclusions more general. Based on current literature, how distinct are vacant lots from urban green space (lawns) etc., how do estimates here compare with other model and measured estimates of urban green space hydrologic benefits in other locations?

We agree, and have restructured these sections accordingly. We have removed ~200 words from the discussion section. This section puts the focus on key messages and makes better linkages between commonly-understood concepts.

Abstract 28 - ‘present’ is awkward - present what?

Given the reviewer’s indication of confusion, we have revised these sentences to:
“Cities evolve through phases of construction, demolition, vacancy, and redevelopment, each impacting water movement at the land surface by altering soil hydrologic properties, land cover, and topography. Currently unknown is whether the variable physical and vegetative characteristics associated with vacant parcels and introduced by demolition may absorb rainfall and thereby diminish stormwater runoff.”

line 34 - not sure how ‘comparing pre- and post demolition partitioning’ fits into this sentence, partitioning of what? What are you assuming about pre-demolition here? Re-word for clarity.

To clarify, we have changed this sentence to: “By extending field estimates to 2400 vacant parcels, we estimate vacant lands citywide may cumulatively infiltrate 51-54% additional volume of rainfall annually as compared to pre-demolition state ...”

line 35 - I don’t see how these finding differentiate vacant lots as ‘multi-functional’ landscapes. First the authors have not really made the case that vacant lots are multi-functional - only that that infiltrate water AND it is not clear that other types of land use are not multi-functional as well.

We have updated this sentence to:
“Our findings differentiate vacant lots as purposeful landscapes that can alleviate large water fluxes into aging wastewater infrastructure.”

line 47 - In some places the redevelopment value of vacant land maybe significant....include citation?

We agree that this sentence is out of place, given the burgeoning examples for redevelopment and repurposing of vacant lands. We have therefore removed it from the manuscript.

line 59 - I don't see how rainfall patterns (at the time scale relevant to detention capacity) force surface landform and soil properties

We agree that this sentence is unclear and have updated it to: "In addition to rainfall rate, detention capacity will be modified by surface landform and soil properties, which present as highly variable across small (e.g., parcels) to large (e.g., cities) spatial scales^{17,24}."

line 75 - how is accounting for distributed presence (of what?) a strategy? This wording does not communicate what I think the authors are trying to say

In the US, many cities are under consent orders from the Environmental Protection Agency to reduce combined sewer overflows. These overflows stem from wastewater systems receiving stormwater runoff as well, thereby reducing system capacity. One way to approach this predicament is to implement stormwater control measures that keep excess stormwater volume out of the wastewater collection and conveyance system. Variants of green infrastructure can be used to absorb and otherwise detain rainfall, and with proper design, prevent return flow through the subsurface back into the system. Vacant lots are presently recognized as a type of green space that functions as a stormwater control measure (Herrmann et al., 2018). To convey this, we have updated this sentence to:

Lines 74 – 81: "An emergent management strategy in the USA is to employ permeable vacant lands as a way to reduce the generation of excess stormwater runoff⁴². This potential runoff is detained in the vacant parcels and kept out of the wastewater collection and conveyance system. In this way, vacant land may be used to reduce the frequency and volume of CSOs, and help fulfill the objectives of federally-mandated consent orders and their associated long-term control plans. This strategy is integrated into the demolition process, which creates vacant land, transforming formerly impervious to pervious surfaces that can be used as a structural tool in stormwater management."

line 79 - you can't really posit that cities have large portfolios (you aren't testing this posit)- and some cities don't have large areas of vacant lots- Rather in cities that do have large portfolios of decentralized pervious surfaces there are opportunities.

This is a great suggestion for rephrasing this point. We have updated this sentence to: "In this vein, we posit that vacant lands may represent an opportunity for managing urban stormwater in cities with large portfolios of permeable vacant parcels that are decentralized throughout a city (and its wastewater service areas)^{15,26}."

line 81 - I'm not sure what is meant by footprint here - this implies that the amount of urban vacant land is growing and I don't see evidence of this in your background

We agree that this implies current change when in fact we are referring to historical change. We have therefore updated this to: "This footprint of urban vacant land with potentially high

detention capacity exists in many cities that are simultaneously struggling to manage excessive stormwater inputs...”

line 75-87 - this is a key paragraph for setting up the background - but needs some more thinking to make sure that the authors do not over state the case - and the meaning of sentences is clear...

We agree– based on the reviewer’s earlier feedback, we have revised the first three sentences of this paragraph to:

“An emergent management strategy in the USA is to employ permeable vacant lands as a way to reduce the generation of excess stormwater runoff⁴². This potential runoff is detained in the vacant parcels and kept out of the wastewater collection and conveyance system. In this way, vacant land may be used to reduce the frequency and volume of CSOs, and help fulfill the objectives of federally-mandated consent orders and their associated long-term control plans. This strategy is integrated into the demolition process, which creates vacant land, transforming formerly impervious to pervious surfaces that can be used as a structural tool in stormwater management.”

line 95 - at this point in the paper, the narrative switches to a single city - Buffalo. An analysis using a single study site is fine - but it needs some introduction - why is Buffalo a good place to do this analysis? What characteristics of this city make it a good example of the kind of potential vacant land benefits that have been discussed thus far.

This is a good point. We have updated this to: “Buffalo is one of many so-called shrinking cities where massive population declines have left many properties vacant or abandoned, and where citywide efforts have been devoted to reducing vacancies through demolition^{2,5,8,9}.”

line 101 This model is central to your analysis - a bit more upfront description here would help readers understand your results - just 1-2 sentences.

To clarify our approach, we have updated this to: “To document the potential benefit of urban vacant parcels for stormwater management, we applied a simple infiltration-excess model. This model partitioned precipitation inputs between infiltration and runoff based on averaged, measured parcel infiltration rates (estimated as maximum infiltration rates) and pervious lot area. We applied this model and a synthesis of field observations towards four key goals...”

line 103 - what is meant by a ‘set’ here - rainfall events in sequence?

This has been updated to “...to determine thresholds of storm magnitude and intensity that initiate runoff from these properties...”

line 107-108 - how are vacant lands evolving? (implies the vacant lands are somehow changing?) - it is also unclear what contextualized means here - and what is meant by a recent building footprint - recent in what way?

The evolution we are referring to are the cycles of building, demolition, and re-development that are ongoing in many cities around the world.

To clarify this, we have updated this sentence to: "...document the evolution of vacant lands (including the effects of redevelopment) as landscape-level stormwater control measures, and – critically – quantifying the impact of these measures on hydrologic partitioning at the land surface."

line 122 - - 'that' has an unclear antecedent - rephrase

This has been revised to:

"We found no evidence linking variations in demolition technique (e.g., how contractors approached the demolition process, sourced soil backfill material) with measured infiltration rates (period: 2001-2013, Fig. S10, Table S8)."

line 127 - Some discussion of why infiltration rates can be a reasonable proxy for detention capacity is needed here - a capacity is typically a maximum potential storage value So there are two points here that need justification 1) is it reasonable to assume that all infiltration is retained as storage (rather than lost through drainage) and 2) presumably infiltration rate varies with rainfall intensity - a capacity suggests a maximum value - so clarify if maximum infiltration rates used here. If so how were they determined.

This is a great point. We have clarified this sentence to:

"We estimated rainfall detention capacity, expressed in terms of flux (length per unit time), as a function of rainfall rate, a maximum infiltration rate, and areal extent of pervious surfaces, all of which will determine the capacity of these parcels to absorb rainfall. Maximum infiltration rates were determined from field observations as unsaturated hydraulic conductivity ($K(-2 \text{ cm})$)."

More details on how these rates were determined are described in the Methods section, under 'Vacant parcel assessment'.

We are essentially imposing a surface capacity (really, a rate) to describe an infiltration capacity. This, of course, has limitations, which are discussed on page 18-19 lines 350 – 369. These limitations are also discussed in the response to Reviewer 1 above, regarding both SSURGO soil databases and our added text to contextualize observations of water table depths.

Nevertheless, we (and most city databases) do not have information regarding the depth to bedrock in these areas, and we have limited information on how soil properties vary with depth. Regardless of the extent to which these characteristics matter for estimating the amount of water infiltrated in vacant lots, our first-order approximation is a strong starting point for quantifying and valuing hydrologic services in these areas.

Fraction of precipitation that is infiltrated is much more clearly defined (e.g as shown in Figure 2 - using the term 'rainfall detention capacity' adds confusion –

We see ‘rainfall detention capacity’ as a broad term that describes volume and depth, and allows us to centralize terminology for ecosystem services associated with vacant parcels. This term is defined in the second paragraph of the introduction (lines 56 – 58). We have retained its use throughout the manuscript, but attempt to show this value in easily-interpretable ways (e.g., fraction of precipitation infiltrated in Figure 2, depth of water infiltrated per vacant lot area in Figure 4).

line 144-147 this explanation is hard to follow - what does it mean to employ infiltration rate across pervious fractions - How is parcel ‘condition’ summarized (what is condition)

We have updated this to: “To derive these estimates, we summarize parcel hydrologic condition using infiltration rates applied to pervious lot fractions.”

line 146 - what is the ‘infiltration-excess processes’ referred to here - description of the approach needs to be more precise - Equations would be helpful

We have updated this sentence to: “We partitioned rainfall using observed infiltration rates as a maximum value above which any precipitation would become direct runoff (generated by infiltration-excess) and below which any precipitation was assumed to be infiltrated into the subsurface.”

To maintain the style of this journal, we retain all equations that are included in the Methods (lines 467 – 482).

line 156-157- how is uncertainty computed? (e.g uncertainty due to what?)

This has been updated to: “Using a subset of the data ($n = 426$), we estimated the impact of measurement uncertainty (1 cm^3) on the fraction of precipitation that would infiltrate into parcels across different hourly design storms.”

line 161 aggregated over what?

We have updated this to: “Across all parcel-averaged infiltration rates, our calculations indicate...”

line 182 - what is meant by the ‘the greater extent of the parcel’ greater than what?

This has been updated to: “...minimal soil compaction over a majority of each parcel...”

line 177 include the statistical method that was used to identify relationships between infiltration rates and characteristics noted.

We explore correlations between infiltration rate and parcel characteristics in Supporting Information (Section S8). This analysis was performed to determine whether the parcels fit the common conceptual model of how precipitation interacts with vacant parcels. We therefore do not state a specific statistical method at this point in the text.

line 196 - was it slow that averaged below 1.5%?

This has been updated to: “Nearly 88% of lots drained toward the street; average slopes across all parcels were less than 1.5% (Fig. 3).”

line 200-203 while I agree that all of these features likely matter (and most simple conceptual models of hydrologic response) would agree, this discussion is overly vague. How does the gentle slope increase water detention (I would expect the opposite)

To clarify our intent and reasoning, we have updated this sentence to: “We expect the gentle slope of the land surface to slow runoff velocities (compared to steeper slopes), allowing greater contact time for runoff to pond at the surface and either infiltrate or evaporate. This assumes that a greater fraction of precipitation than estimated via our simple modeling may be retained on these parcels, and not routed via the curb-gutter-street network to storm sewer inlets.”

A key question is how runoff is partitioned between surface detention store and transpiration fluxes that lead to water returning to the atmosphere versus drainage - either rapid or shallow subsurface - It would be helpful to provide some insight into this

We agree that this is a key question, but one beyond the scope of our analysis and modeling efforts. Evapotranspiration – as combined evaporation and transpiration – can remove water from these areas, mostly by reducing near-surface and rooting zone soil moisture content. Incorporating these processes will likely lead to even greater detention capacities than we have estimated here. We provide discussion of evapotranspiration and deep drainage on pages 18-19 lines 343 – 369.

Figure 3 - how specifically do these features shape infiltration (e.g what analysis was done to support this statement)

This has been updated to (lines 229 – 232): “...will interact with infiltration rate (n = 520, see Table S2) to favor infiltration versus runoff on vacant parcels.”

line 224 - is there data suggesting that vacant land in Buffalo is expanding?

Vacant land heavily expanded from 2001 through 2013. However, we agree that this phrase may dilute the overall point of our statement. We have clarified this sentence to: “When taken together, the characteristic internal structure of assessed parcels suggests that vacant land in Buffalo is a hydrologic sink for rainfall.”

line 228 ‘representative assessment of average pace hydrology’ is not clear -explain what this is - do you mean average maximum infiltration rate?

We did not apply an average maximum infiltration rate across all sites. To clarify how we used random resampling, we have updated this to: “To further confirm this, we randomly resampled

our representative assessment of average parcel hydrology (n = 520) across a larger dataset of unassessed vacant parcels to calculate cumulative, city-wide rainfall detention (Fig. 4).”

These methods are described on lines 526 – 532: “Our analysis for 520 properties was extended to 2482 sites with known parcel areas and delineated building footprints. We randomly resampled the distribution of infiltration rates from 520 properties to these 2482 properties 1000 times, and subsequently calculated *RDT* across warm-season precipitation from a twenty-one-year record. We assumed a maximum rate of infiltration at the surface for each timestep. Fig. 4 displays *RDT* across a concentrated period of blight control and initial redevelopment, demonstrating the cumulative impact of demolition (Fig 4). Results for annual precipitation regimes (1997 – 2017) are tabulated in SI (Section S12).”

line 230 - provide some indication of how this estimate of footprints was made - using what data and assumptions (and time period over which removal occurred)

We have clarified this to:

“Using historical imagery from 2002, our conservative estimates of former building footprints suggest that on the order of 0.29 km² of pervious area was gained across a combined 2482 lots.”

We have also clarified this in the methods (lines 476 – 483):

“Pre-demolition, post-demolition, and redevelopment states

Aerial imagery across Buffalo was leveraged to hand-delineate former building footprints on 2482 vacant parcels for two time points (2002, 2017). Orthoimagery from 2002 was used to establish a pre-demolition estimate of former impervious cover, assumed to be an underestimate given that many parcels also included sidewalks, disconnected garages, and sheds. Sites demolished prior to 2002 were excluded from analysis. Imagery from 2017 was used to establish a more recent estimate of building footprints (assumed to be a good approximation for impervious cover) on vacant properties, accounting for incomplete demolition or redevelopment.”

line 233 how were rates of redevelop and demolition determined?

We have clarified this line to: “Accounting for redevelopment and incomplete demolition (ca. 2017), we estimate approximately 0.0028 km² of impervious surface across these areas; this estimate may continue to grow with ongoing redevelopment.”

These were determined by delineation of building footprints on 2482 vacant parcels. As stated in the methods (lines 476 – 483):

“Pre-demolition, post-demolition, and redevelopment states

Aerial imagery across Buffalo was leveraged to hand-delineate former building footprints on 2482 vacant parcels for two time points (2002, 2017). Orthoimagery from 2002 was used to establish a pre-demolition estimate of former impervious cover, assumed to be an underestimate given that many parcels also included sidewalks, disconnected garages, and sheds. Sites demolished prior to 2002 were excluded from analysis. Imagery from 2017 was used to establish a more recent estimate of building footprints (assumed to be a good approximation for

impervious cover) on vacant properties, accounting for incomplete demolition or redevelopment.”

line 246 - clarify what the baseline is here - increased relative to what?

We have clarified this to: “Our analysis indicates that demolition of buildings on ca. 2400 properties (comparing pre- to post-demolition) has increased the cumulative rainfall detention rate by an average of 52% (range: from 51 to 54%) across the city (Fig 4; Table S12).”

line 267 - it would helpful to be more precise here - what data is used to estimate the thousands of parcels?

Our estimate of the number of parcels that are vacant across the City of Buffalo comes from the City’s property inventory, which is cited in the Data Availability statement.

line 264 - I don’t really understand the point here - if building footprints are small - then vacancy many not significantly add to expected infiltration - e.g if imperviousness is small

We agree that a few points were being mixed in this paragraph. We have clarified this paragraph to the following (297 – 303): “In Buffalo, the expanse of vacant land disconnects or eliminates a substantial amount of connected impervious surface (~34% of total vacant land area; Figure 4b). Altogether, this creates a comparatively large area of hydraulically-disconnected land mass with characteristics that favor detention (Figure 4c). Considering that our estimates are based on 2482 parcels representing only a fraction of the thousands of parcels that have been demolished across the city (Fig. S1), our assessment represents an underestimate of the overall benefits from demolition, even when considering redevelopment.”

Could the authors provide a simple estimate of the range in pervious cover associated with developed parcels - and then use that quantitative estimate in the comparisons?

I think the authors have done this- but it is challenging to extract this information from the paper as written

The range of pervious cover associated with developed parcels is 3% to 100%, which shows just how variable of an area building footprints can occupy. Though the average percentage of impervious cover on formerly developed parcels is already stated on line 249, we have added this again on line 262.

line 277 - this was true ONLY in Buffalo - The sentence implies a generality that is not there, So in Buffalo, we found that...

This has been updated to: “...we found that almost 60% of parcels in Buffalo had infiltration rates greater than the 75th percentile of historical hourly rain intensity...”

line 293-295 there are other studies that document the role of vegetated patches in urban environments on reducing urban runoff - it would be useful to place estimates found her

into this broader literature context.

We agree that broadening our findings within the context of other literature on the impacts of urban green spaces on hydrological processes/partitioning would be helpful at this point in the manuscript and in other places. Based on this recommendation, we have made the following changes to the text:

Page 17-18 lines 331 – 334 “Our observations that greened vacant parcels may favor infiltration and reduce runoff place these urban green spaces in the context of a larger body of literature, which documents runoff reduction via green infrastructure⁴⁷⁻⁴⁹, lawns⁵⁰⁻⁵², and street trees⁵³⁻⁵⁴ across cities.”

Page 21 lines 414 – 420: “Though the stormwater capacities of residential lawns have been widely investigated^{51,64-65}, we currently have little to no baseline for comparison regarding how infiltration rates on vacant lots compare with nearby residential lawns. Field studies that have linked construction date (as a proxy for last date of soil disturbance) to infiltration rates on residential lawns suggest that if properly maintained, many residential green spaces are likely to provide additional capacity for stormwater volume, beyond those attributed to vacant lands^{51,64-65}.”

line 314 - what may be underestimated?

This has been updated to: “... the presence of vegetation on vacant parcels may amplify the rainfall detention capacity of these areas. This suggests that this detention capacity may be underestimated in our analysis.”

line 317 - meaning of landscape or system “storage versus their respective capacities” is unclear

This sentence has been removed from our analysis due to restructuring of the Discussion section.

line 335 this is too vague and mixes concepts of sustainability and equity - without meaningfully describing the relationship with either concept

We have updated this sentence to clarify that equity is intertwined with the lifecycle of vacant lots. This new sentence reads:

“Research is also needed to further disentangle the complex interactions between human-derived and natural landscape dynamics that regulate how vacant lot hydrologic benefits are rendered. The evolution of vacant lands must be balanced with social equity, maintenance of civic services, and stabilization of real estate values at the neighborhood level⁶⁰.”

REVIEWERS' COMMENTS:

Reviewer #1 (Remarks to the Author):

The authors have responded to my additional questions, added appropriate material and clarified uncertain areas. I think this is ready for publication and look forward to seeing this in press.

Reviewer #2 (Remarks to the Author):

The authors have now addressed my concerns... I appreciate the detailed responses. The manuscript reads well and I think it will make a valuable , interesting contribution to the literature

Christina Tague

Reviewer #1 (Remarks to the Author):

The authors have responded to my additional questions, added appropriate material and clarified uncertain areas. I think this is ready for publication and look forward to seeing this in press.

We appreciate your feedback.

Reviewer #2 (Remarks to the Author):

The authors have now addressed my concerns... I appreciate the detailed responses. The manuscript reads well and I think it will make a valuable , interesting contribution to the literature

Christina Tague

We appreciate your feedback.